# KOALA++: Efficient Kalman-Based Optimization with Gradient-Covariance Products

**Zixuan Xia**[*]
zixuan.xia@students.unibe.ch

**Aram Davtyan**[†]
aram.davtyan@unibe.ch

**Paolo Favaro**[†]
paolo.favaro@unibe.ch

## Abstract

We propose KOALA++, a scalable Kalman-based optimization algorithm that explicitly models structured gradient uncertainty in neural network training. Unlike second-order methods, which rely on expensive second order gradient calculation, our method directly estimates the parameter covariance matrix by recursively updating compact gradient covariance products. This design improves upon the original KOALA framework that assumed diagonal covariance by implicitly capturing richer uncertainty structure without storing the full covariance matrix and avoiding large matrix inversions. Across diverse tasks, including image classification and language modeling, KOALA++ achieves accuracy on par or better than state-of-the-art first- and second-order optimizers while maintaining the efficiency of first-order methods. **The code is publicly available at** https://github.com/Sumxiaa/KOALA_Plus_Plus.

## 1 Introduction

Optimization lies at the heart of deep learning. As neural networks continue to scale in depth, width, and data complexity, the efficiency, stability, and scalability of optimization algorithms have become increasingly important. First-order methods such as stochastic gradient descent (SGD) [28] and its adaptive variants like Adam [17] and RMSProp [30] offer practical efficiency and have driven much of the progress in deep learning. However, they often suffer from instability, slow convergence, and sensitivity to hyperparameter tuning – particularly in large-scale or noisy training regimes [16, 32].

To mitigate these issues, second-order optimizers such as K-FAC [20] and Shampoo [8] introduce curvature-aware updates by approximating the Hessian or Fisher information matrix. These methods often improve convergence and generalization, but require costly matrix operations, limiting their scalability. Recently, AdaFisher [7] has emerged as a more scalable alternative, using block-diagonal Kronecker approximations of the Fisher information matrix to reduce overhead while preserving second-order benefits. Despite its effectiveness, AdaFisher fundamentally depends on the quality of the Fisher matrix estimation, which remains sensitive to noise and mini-batch variability.

An alternative and theoretically grounded direction comes from Kalman filtering. In Bayesian estimation theory, the posterior covariance matrix of an unbiased estimator defines the *tightest achievable lower bound* on parameter uncertainty, as formalized by the Cramér–Rao bound [24, 3]. Although the Fisher information matrix serves as an *inverse bound* proxy, it does not directly capture the true uncertainty in the parameters. Kalman filters, on the other hand, propagate the full covariance matrix of the estimate through time, providing a more faithful representation of uncertainty – particularly in noisy or nonstationary environments.

---

[*]Work done during Master studies at the University of Bern

[†]Computer Vision Group, University of Bern – https://www.cvg.unibe.ch

39th Conference on Neural Information Processing Systems (NeurIPS 2025).

Motivated by this insight, KOALA [4] proposed to frame stochastic optimization as a Kalman filtering problem, which models parameter updates as recursive state estimation, using a covariance matrix to adjust step sizes adaptively. However, KOALA's original formulation relies on overly simplistic approximations, such as diagonal covariances and idealized gradient assumptions, which limit its practical effectiveness.

In this paper, we introduce **KOALA++**, a computationally efficient extension of the KOALA framework that explicitly maintains structured gradient covariance information. **KOALA++** tracks the evolution of uncertainty directly through a recursive update at each training iteration $k$ of the gradient-covariance product $v_k = H_k P_{k-1}$, which we call *surrogate*, where $H_k$ is the gradient of the loss function and $P_{k-1}$ is the prior estimate of the parameter covariance. The key idea in our method is to approximate the recursive gradient-covariance update so that it never uses the covariance matrix $P_{k-1}$. By doing so, **KOALA++** effectively avoids expensive matrix inversions and storing the entire covariance matrix $P_{k-1}$, while benefiting from the lifted constraints on its structure. Moreover, **KOALA++** retains scalability akin to first-order methods, while achieving optimization dynamics and final performance competitive with state-of-the-art second-order methods.

**Our main contributions are summarized as follows:**

- We propose **KOALA++**, a Kalman-inspired optimizer that explicitly propagates structured gradient uncertainty via covariance modeling.
- We derive an efficient recursive update for the surrogate $v_k$ based on a minimum-norm solution to the covariance estimation problem.
- We implement **KOALA++** with minimal overhead and validate its effectiveness across diverse architectures and tasks, including the ResNet family [9], vision transformers [18], and language models.
- Extensive experiments show that **KOALA++** consistently matches or exceeds the accuracy of first- and second-order baselines while offering improved stability and a better efficiency/performance trade-off.

## 2 Related Work and Motivation

### 2.1 Optimization Methods: A Brief Overview

First-order optimizers such as SGD [28] and Adam [17] are widely adopted for their scalability and simplicity, but often ignore the structure of gradient noise and require manually tuned learning rate schedules.

Second-order optimization methods, including K-FAC [20], Shampoo [8], and more recently AdaFisher [7], seek to address these shortcomings by explicitly incorporating curvature information. Nevertheless, practical implementations rely heavily on fixed approximations, such as block-diagonal or Kronecker-factored representations of the Hessian or Fisher matrices, to maintain computational feasibility. However, such approximations either still show limitations in terms of efficiency, when compared to first order methods, or adopt drastic simplifications that make the estimation process less stable.

### 2.2 Optimization via Kalman Filtering

The idea of using Kalman filtering for optimization has a long history. Early works include scalar Kalman updates for stochastic optimization [13], applications in reinforcement learning [29], and connections between the Extended Kalman Filter (EKF) and natural gradient methods [22]. Although these approaches demonstrate the versatility of Kalman-based updates, they are often task-specific or limited to small-scale models, and computing full covariance updates remains prohibitively expensive.

More recent research has revisited the Kalman filtering perspective in the context of large-scale neural optimization. Duran-Martin et al. [6] and Chang et al. [2] interpret learning as a Bayesian state estimation problem, where the network parameters are treated as latent states, and the likelihood of the data provides an explicit probabilistic observation model. These methods apply low-rank or subspace-structured versions of the Extended Kalman Filter to maintain tractable posterior updates.

Their formulation operates in the full parameter or feature space (*i.e.*, with matrix-valued state and observation models), which implies a computational cost of $\mathcal{O}(n^2)$ to $\mathcal{O}(n^3)$ per update, where $n$ is the number of model parameters, even under low-rank approximations. Consequently, they are mainly applicable to some relatively small scale settings.

In contrast, KOALA [4] reinterprets gradient-based optimization as a scalar state estimation problem, where the loss function itself serves as an implicit observation. This design removes the need for an explicit probabilistic target and allows Kalman-style uncertainty propagation to be applied directly to general stochastic objectives. The formulation therefore scales linearly with the number of parameters, achieving $\mathcal{O}(n)$ complexity in practice, while remaining compatible with arbitrary differentiable loss functions. This implicit observation model makes KOALA more general and applicable to a wider range of optimization problems than Kalman filters formulated on explicit probabilistic observations. Nevertheless, the original KOALA formulation still involves certain simplifying assumptions that limit its expressiveness and stability in complex, anisotropic optimization landscapes.

For comparison, SLANG [21] focuses purely on efficient covariance estimation through the diagonal plus low-rank structure without invoking any Kalman filtering or Bayesian state-update interpretation. Although SLANG achieves improved posterior approximations compared to diagonal methods, its updates are not dynamically informed by temporal prediction-correction as in Kalman filtering, and its computational cost remains at least $\mathcal{O}(n^2)$.

In general, these methods differ in their dimensionality, observation modeling, and scalability. A gap remains between the expressiveness of structured covariance estimation and the efficiency of scalar Kalman filtering. This motivates the design of **KOALA++**, which preserves KOALA's first-order scalability while introducing directionally aware covariance updates for large-scale optimization.

### 2.3 Motivation for KOALA++: Towards Structured Covariance Updates

The original KOALA formulation addresses efficiency by interpreting optimization as a scalar Kalman filtering problem, where the loss function acts as an implicit observation. However, its covariance update assumes $P_k \approx \sigma^2 I$ and substitutes $H_k^\top H_k$ with gradient norms, which may reduce some information to a single scalar, limiting its ability to adapt to anisotropic curvature or correlated gradient directions, particularly in deep or noisy regimes.

On the other hand, as discussed in subsection 2.2, better approximations of the covariance matrix can provide richer uncertainty representations, but they are computationally prohibitive. This trade-off leaves a gap between expressiveness and scalability that current methods have yet to bridge.

Thus, **KOALA++** is designed to overcome this limitation while retaining the first-order scalability of KOALA. Rather than maintaining or constraining a full covariance matrix, **KOALA++** explicitly tracks a *directional covariance projection*

$$v_k := H_k P_{k-1},$$

which quantifies how uncertainty evolves along current gradient directions. This projection is updated via an approximate recursive least-squares step that implicitly captures curvature anisotropy without requiring explicit matrix storage or inversion. As a result, **KOALA++** provides a lightweight mechanism for propagating structured uncertainty, offering an effective balance between expressiveness and computational efficiency.

## 3 KOALA++

### 3.1 Problem Formulation and Kalman Filtering Notation

Neural network training is fundamentally an optimization problem, where, given a dataset $\mathcal{D} = \{(x_i, y_i)\}_{i=1}^N$, the objective of the learning task is to adjust the model's parameters $\theta \in \mathbb{R}^n$ so that the network's predictions $f(x_i; \theta)$ minimize the empirical risk

$$\min_\theta \mathcal{L}(\theta), \quad \text{with } \mathcal{L}(\theta) = \frac{1}{N} \sum_{i=1}^N l(f(x_i; \theta), y_i), \tag{1}$$

where $l$ is a chosen loss function. However, in modern deep learning applications, $N$ is often large, which makes full-batch optimization computationally infeasible. As a result, practitioners rely on

Table 1: EKF equations from [4].

| | |
|---|---|
| $S_k = H_k(P_{k-1} + QI)H_k^\top + R$ | innovation covariance, where $H_k = \nabla L_k^\top(\theta_{k-1})$ |
| $K_k = (P_{k-1} + QI)H_k^\top S_K^{-1}$ | Kalman gain |
| $P_k = (I - K_k H_k)(P_{k-1} + QI)$ | posterior of the state update |
| $\theta_k = \theta_{k-1} + K_k(L_k^{\text{target}} - L_k(\theta_{k-1}))$ | state estimate update |

stochastic optimization methods that work on minibatches of data and minimize estimates of the loss

$$L_k(\theta) = \frac{1}{m} \sum_{i \in B_k} l(f(x_i; \theta), y_i), \tag{2}$$

where $B_k$ is a set of indices in the $k$-th minibatch and $m = |B_k|$ is the cardinality of the minibatch. Although minibatch-based approaches such as SGD [28] and its adaptive variants are effective and scalable, they introduce noise due to the stochastic nature of loss and gradient estimates.

To mitigate this issue, a recent approach, KOALA [4], casts neural network training as a recursive state estimation problem using Kalman filtering [15], a principled Bayesian framework for recursively estimating the hidden state of a system from noisy observations. At each iteration $k$, it combines prior estimates with new measurements to update the state $\theta_k$ and its associated uncertainty in the form of a covariance matrix $P_k$. Importantly, by keeping track of an estimate of the system's state's covariance, Kalman filtering is essentially implicitly using second order information about the state variables. The standard Extended Kalman Filter (EKF) generalizes this to non-linear systems using local linearization. In KOALA, model parameters $\theta_k$ are treated as latent states with stationary evolution under process noise, and the minibatch loss function $L_k$ serves as a noisy observation of a target loss value $L_k^{\text{target}}$. In these settings, the EKF equations can be summarized as in Table 1. To remain tractable in high-dimensional settings, KOALA simplifies the EKF update by:

- Assuming an identity-scaled covariance matrix $P_k \approx \sigma_k^2 I$;
- Approximating $H_k^\top H_k$ with $\|H_k\|^2 I$;
- Reducing all matrix operations to scalar computations.

The resulting update closely resembles SGD, but with an adaptively scaled learning rate governed by the uncertainty estimate $P_k$.

While KOALA demonstrates promising stability and simplicity, its reliance on scalar approximations limits its capacity to capture directional or structured noise in high-dimensional loss landscapes. To address this, we propose **KOALA++**, which enhances the original framework with an efficient estimation of the directional state covariance while preserving the Kalman-based update algorithm.

### 3.2 Kalman Update with $v_k$-Based Reparameterization

To overcome the limitations of isotropic covariance assumptions in KOALA, **KOALA++** introduces a more expressive formulation that captures directional uncertainty without incurring the full cost of matrix-valued updates. Instead of updating the full posterior covariance $P_k \in \mathbb{R}^{n \times n}$, we propagate a low-rank surrogate

$$v_k := H_k P_{k-1} \in \mathbb{R}^{1 \times n}, \tag{3}$$

which represents the projection of the parameter covariance along the current measurement direction. This allows **KOALA++** to maintain a richer understanding of uncertainty and curvature structure while keeping the computational complexity comparable to first-order methods.

**Low-Rank Reparameterization of the Kalman Update:** By assuming a constant process noise $Q_k = Q$ for all $k$, the innovation covariance $S_k$ becomes

$$S_k = H_k(P_{k-1} + Q)H_k^\top + R = H_k v_k^\top + H_k Q H_k^\top + R. \tag{4}$$

Using this expression, the Kalman gain becomes

$$K_k = \frac{(P_{k-1} + Q)H_k^\top}{S_k} = \frac{v_k^\top + Q H_k^\top}{S_k} \tag{5}$$

and the corresponding state update is

$$\theta_k \leftarrow \theta_k + \left(L_k^{\text{target}} - L_k(\theta_k)\right) \frac{v_k^\top + QH_k^\top}{S_k}. \tag{6}$$

Notice that, as suggested in [4], the learning rate of the update can be controlled by selecting a specific $L_k^{\text{target}}$. For example, $L_k^{\text{target}} = (1 - \eta_k)L_k(\theta_{k-1})$ ensures that the update is scaled by $\eta_k$.

**Recursive Update of $v_k$:**   To avoid storing $P_{k-1}$ explicitly, we recursively update $v_k$ by substituting the Kalman covariance update $P_{k-1} = (I - K_{k-1}H_{k-1})(P_{k-2} + Q)$

$$
\begin{aligned}
v_k = H_k P_{k-1} &= H_k(I - K_{k-1}H_{k-1})(P_{k-2} + Q) \\
&= \left(H_k - H_k \frac{v_{k-1}^\top + QH_{k-1}^\top}{S_{k-1}} H_{k-1}\right)(P_{k-2} + Q) \\
&= H_k P_{k-2} + H_k Q - H_k \frac{v_{k-1}^\top + QH_{k-1}^\top}{S_{k-1}} v_{k-1} - H_k \frac{v_{k-1}^\top + QH_{k-1}^\top}{S_{k-1}} H_{k-1} Q.
\end{aligned}
\tag{7}
$$

For simplicity, we define the recurring term

$$\lambda_k \doteq H_k \frac{v_{k-1}^\top + QH_{k-1}^\top}{S_{k-1}}, \tag{8}$$

which allows us to rewrite the update for $v_k$ in a more compact way:

$$v_k = H_k P_{k-2} - \lambda_k v_{k-1} + (H_k - \lambda_k H_{k-1})Q. \tag{9}$$

In the last equation, all terms except the first can be efficiently calculated in iteration $k$. The first term cannot be easily evaluated due to the term $P_{k-2}$. The objective is to never store it explicitly to avoid the quadratic storage demand. As a first attempt, one could approximate $H_k$ with $H_{k-1}$ and replace the problematic term with $v_{k-1}$. However, we observed experimentally that this does not work well because of the high variability of the minibatch gradients.

As an alternative, we aim to retrospectively approximate the prior covariance matrix $P_{k-2}$ instead. We do this by taking advantage of the relation $v_{k-1} = H_{k-1}P_{k-2}$. However, since this equation is generally underdetermined in high-dimensional settings, we formulate a regularized inverse problem to obtain a unique and tractable solution. Specifically, we adopt the minimum Frobenius-norm formulation

$$\min_{P_{k-2}} \|P_{k-2}\|_F^2 \quad \text{subject to} \quad H_{k-1}P_{k-2} = v_{k-1}, \tag{10}$$

which yields a closed-form least-squares solution. This construction allows us to approximate $P_{k-2}$ efficiently without explicitly storing or inverting full covariance matrices.

For computational and memory efficiency, we use only one linear constraint, $v_{k-1} = H_{k-1}P_{k-2}$, although more could be used in principle. However, introducing more constraints, *e.g.*, projections of other previous gradients $H_{k-2}, H_{k-3}$, etc., would also require memorizing more gradient terms and using more calculations to update $v_k$. The use of the minimum Frobenius-norm criterion is to ensure a stable update for $v_k$ by limiting the norm of $P_{k-2}$.

In the next subsection, we discuss two variants of our algorithm, depending on the nature of the solution to the optimization problem (10).

### 3.3   Least-Squares Estimate of $P_{k-2}$

**Variant I: Vanilla Least-Squares Estimation**   In the first variant of our approach, we solve the least-squares problem without imposing any structural constraints on the covariance estimate. Given the relation $v_{k-1} = H_{k-1}P_{k-2}$, we compute $P_{k-2} \in \mathbb{R}^{n \times n}$ by introducing the Lagrange multipliers $\mu \in \mathbb{R}^{1 \times n}$ and write the Lagrangian

$$\mathcal{L}(P_{k-2}, \lambda) = \|P_{k-2}\|_F^2 + (H_{k-1}P_{k-2} - v_{k-1})\mu^\top. \tag{11}$$

---
**Algorithm 1** KOALA++

Initialize $\theta_0$, $v_1$, $Q$, $R$, and fix the learning rate schedule $\eta_k$

**for** $k = 2$ to $T$ **do**

    For simplicity, denote $H_k = \nabla L_k^\top(\theta_{k-1})$

    Calculate $\alpha_k, \lambda_k, r_k$ respectively from Equations (13), (8), and (16)

    **Update:**

$$v_k = (\alpha_k - \lambda_k)v_{k-1} + (H_k - \lambda_k H_{k-1})Q + r_k H_{k-1} \tag{19}$$

$$\theta_k = \theta_{k-1} - \frac{\eta_k L_k(\theta_{k-1})}{H_k v_k^\top + H_k Q H_k^\top + R} \cdot (v_k^\top + Q H_k^\top) \tag{20}$$

**end for**

**return** $\theta_T$

---

This formulation admits the closed-form solution[3]

$$P_{k-2} = \frac{H_{k-1}^\top v_{k-1}}{\|H_{k-1}\|^2}. \tag{12}$$

By substituting this solution into $H_k P_{k-2}$ gives

$$H_k P_{k-2} = \alpha_k v_{k-1}, \quad \text{where} \quad \alpha_k := \frac{H_k H_{k-1}^\top}{\|H_{k-1}\|^2}. \tag{13}$$

Finally, we can obtain the following vanilla $v_k$ update

$$v_k = (\alpha_k - \lambda_k)v_{k-1} + (H_k - \lambda_k H_{k-1})Q. \tag{14}$$

**Variant II: Symmetric Covariance Estimation**  To improve stability and theoretical consistency with classical Kalman filtering, we introduce a second variant that enforces symmetry in the approximation of $P_{k-2}$. In particular, the additional regularization term $P_{k-2} = P_{k-2}^\top$ is added to the objective. As in the previous derivation, we obtain a closed-form solution for $P_{k-2}$[4]

$$P_{k-2} = \frac{H_{k-1}^\top v_{k-1} + v_{k-1}^\top H_{k-1}}{\|H_{k-1}\|^2} - H_{k-1} v_{k-1}^\top \frac{H_{k-1}^\top H_{k-1}}{\|H_{k-1}\|^4}. \tag{15}$$

When substituting this into $H_k P_{k-2}$, the only difference between the two variants is the term

$$r_k = \frac{(H_k v_{k-1}^\top)(H_{k-1} H_{k-1}^\top) - (H_{k-1} v_{k-1}^\top)(H_k H_{k-1}^\top)}{\|H_{k-1}\|^4}. \tag{16}$$

To simplify notation and implementation, we adopt a unified update rule for both variants. In the unconstrained case, we simply set $r_k = 0$, effectively reducing the update to the asymmetric version. This allows the general form of the update to be written compactly as

$$\boxed{v_k = (\alpha_k - \lambda_k)v_{k-1} + (H_k - \lambda_k H_{k-1})Q + r_k H_{k-1}.} \tag{17}$$

To maintain consistency with the original KOALA formulation and to simplify the calculation in the early stages of training, we initialize the directional covariance product $v_1 = H_1 P_0$ using a scaled identity assumption on the initial covariance matrix $P_0$. Specifically, we set:

$$v_1 := \sigma_0^2 H_1, \tag{18}$$

where $\sigma_0^2$ is a scalar hyperparameter representing the initial uncertainty. This corresponds to assuming $P_0 = \sigma_0^2 I$, which aligns with the isotropic prior used in KOALA and allows us to avoid explicit matrix computations at initialization. The general procedure of **KOALA++** is summarized in Algorithm 1.

---

[3]See Appendix B.1 for a detailed derivation.

[4]See Appendix B.2 for a detailed derivation.

Table 2: Experimental results on CIFAR-10 and CIFAR-100 with different optimizers for 100 and 200 epochs. Best (blue) and second best (orange) performances are in boldface and with shading.

| Dataset | Architecture | Method | Error | | | |
|---|---|---|---|---|---|---|
| | | | 100-epochs | | 200-epochs | |
| | | | Top-1 Err | Top-5 Err | Top-1 Err | Top-5 Err |
| CIFAR-10 | ResNet-18 | SGD | $\mathbf{5.69}_{0.19}$ | $\mathbf{0.18}_{0.03}$ | $\mathbf{4.83}_{0.20}$ | $\mathbf{0.15}_{0.03}$ |
| | | Adam | $6.96_{0.06}$ | $0.27_{0.05}$ | $6.35_{0.13}$ | $0.27_{0.04}$ |
| | | KOALA-M | $6.29_{0.04}$ | $0.37_{0.02}$ | $6.06_{0.14}$ | $0.28_{0.06}$ |
| | | **KOALA++(Ours)** | $\mathbf{5.61}_{0.10}$ | $\mathbf{0.26}_{0.04}$ | $\mathbf{5.43}_{0.19}$ | $\mathbf{0.24}_{0.02}$ |
| | ResNet-50 | SGD | $6.61_{0.12}$ | $\mathbf{0.17}_{0.02}$ | $\mathbf{5.38}_{0.10}$ | $\mathbf{0.14}_{0.05}$ |
| | | Adam | $\mathbf{6.38}_{0.12}$ | $0.26_{0.02}$ | $5.77_{0.13}$ | $0.21_{0.02}$ |
| | | KOALA-M | $8.09_{0.16}$ | $0.29_{0.04}$ | $7.78_{0.39}$ | $0.32_{0.06}$ |
| | | **KOALA++(Ours)** | $\mathbf{5.28}_{0.20}$ | $\mathbf{0.21}_{0.04}$ | $\mathbf{4.90}_{0.21}$ | $\mathbf{0.18}_{0.06}$ |
| | W-ResNet-50-2 | SGD | $6.19_{0.35}$ | $\mathbf{0.17}_{0.03}$ | $\mathbf{4.91}_{0.14}$ | $\mathbf{0.13}_{0.05}$ |
| | | Adam | $\mathbf{6.03}_{0.07}$ | $0.24_{0.03}$ | $5.53_{0.21}$ | $0.27_{0.02}$ |
| | | KOALA-M | $7.62_{0.13}$ | $0.30_{0.04}$ | $7.67_{0.43}$ | $0.32_{0.06}$ |
| | | **KOALA++(Ours)** | $\mathbf{4.94}_{0.25}$ | $\mathbf{0.16}_{0.05}$ | $\mathbf{4.70}_{0.17}$ | $\mathbf{0.17}_{0.01}$ |
| CIFAR-100 | ResNet-50 | SGD | $25.07_{0.88}$ | $6.89_{0.39}$ | $\mathbf{22.28}_{0.31}$ | $\mathbf{5.55}_{0.18}$ |
| | | Adam | $24.25_{0.34}$ | $7.06_{0.15}$ | $24.14_{0.12}$ | $7.47_{0.34}$ |
| | | KOALA-M | $\mathbf{23.71}_{0.33}$ | $\mathbf{6.58}_{0.04}$ | $22.91_{0.38}$ | $6.09_{0.33}$ |
| | | **KOALA++(Ours)** | $\mathbf{22.24}_{0.44}$ | $\mathbf{5.97}_{0.32}$ | $\mathbf{21.86}_{0.06}$ | $\mathbf{5.79}_{0.00}$ |
| | W-ResNet-50-2 | SGD | $23.03_{0.29}$ | $5.97_{0.11}$ | $23.50_{0.15}$ | $6.47_{0.16}$ |
| | | Adam | $23.99_{0.17}$ | $6.69_{0.07}$ | $23.74_{0.19}$ | $7.10_{0.17}$ |
| | | KOALA-M | $\mathbf{21.59}_{0.53}$ | $\mathbf{5.71}_{0.12}$ | $22.22_{0.34}$ | $\mathbf{6.15}_{0.17}$ |
| | | **KOALA++(Ours)** | $\mathbf{21.23}_{0.28}$ | $\mathbf{5.33}_{0.13}$ | $\mathbf{20.33}_{0.21}$ | $\mathbf{5.02}_{0.19}$ |

# 4 Experiments

We evaluate **KOALA++** on a range of vision and language tasks to demonstrate its generality, stability, and convergence behavior compared to existing optimizers. Our experiments are organized into three parts: image classification, language modeling and ablation studies. Due to space constraints, we report a representative subset of results in the main text. Additional evaluations, including ablations, hyperparameter tuning, and extended visualizations, are provided in Appendix E.

## 4.1 Image Classification

**CIFAR-10/100 Classification:** We follow the experimental setup of the original KOALA paper [4] for CIFAR-10 and CIFAR-100 classification tasks, including data augmentation, model architectures, and optimization settings. We introduce two modifications for CIFAR-10: (i) training is repeated with three random seeds (42, 3407, and 2025) to ensure statistical robustness, and (ii) we adopt a two-stage learning rate decay strategy at the 100th and 150th epochs for 200-epoch training (inspired by successful multi-step decay strategies reported in prior works such as [27]), which improves generalization compared to the single-step decay used in the original paper. The results[5] are shown in Table 2.

**Comparison with Recent Advances** To comprehensively evaluate the effectiveness and generalization capability of **KOALA++**, we design two experimental settings that progressively incorporate **more recent optimizers**, **modern network architectures**, **stronger learning rate schedulers**, and **richer data augmentations**. This design mirrors recent benchmarking practices in the literature, particularly those from KOALA [4] and AdaFisher [7].

**(1) CIFAR-100 + ResNet-50 + Advanced Optimizers.** To evaluate **KOALA++** in a standardized yet competitive setting, we follow the benchmarking protocol introduced in KOALA [4], and compare our method with a wide range of recent optimizers on the CIFAR-100 dataset using ResNet-50, trained for 100 epochs. All optimizers are tested under the same learning rate schedule (StepLR) and training configuration. For each optimizer, we adopt hyperparameters as suggested in the original papers or official implementations. The settings and results are shown in Table 3.

---

[5]Some of the earlier reported results were obtained from existing public benchmarks, and therefore mean and standard deviation values are not available for all entries. To ensure reproducibility, we reran the experiments under identical settings; the full code and configurations are available at: `https://github.com/Sumxiaa/KOALA_Plus_Plus`.

Table 3: CIFAR-100 (ResNet-50, 100 epochs, StepLR) results and optimizer hyperparameters.

| Optimizer | Top-1 Err. | Top-5 Err. | Hyperparameters / Source |
|---|---|---|---|
| Yogi [25] | 33.99 | 10.90 | lr=$10^{-2}$, $\beta_1$=0.9, $\beta_2$=0.999, $\epsilon$=$10^{-3}$ |
| Adamax [17] | 32.42 | 10.74 | Same as Adam |
| AdamW [19] | 27.23 | 7.98 | Same as Adam |
| AdamP [10] | 26.62 | 7.61 | Same as Adam |
| Amsgrad [26] | 25.27 | 6.78 | Same as Adam |
| Adan [31] | 24.92 | 6.86 | Paper and repo defaults[6] |
| Fromage [1] | 24.65 | 6.71 | lr=$10^{-2}$ (recommended in GitHub[7]) |
| AdaFisher [7] | 23.38 | 6.05 | Original AdaFisher hyperparams |
| Adabelief [33] | 23.07 | 6.05 | Official config[8] |
| KOALA-M | 23.71 | 6.58 | Redid the experiments with our implementation settings |
| **KOALA++** (Ours) | **22.24** | **5.97** | Based on our implementation settings |

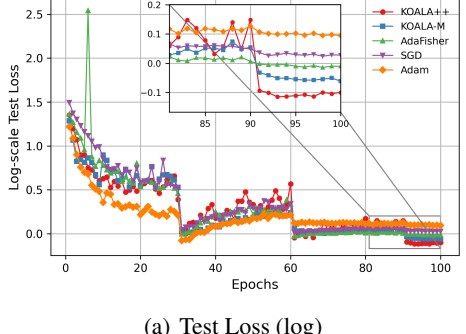

(a) Test Loss (log)

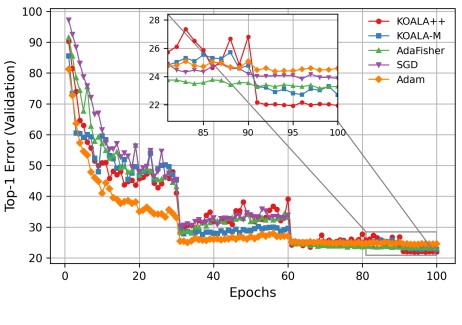

(b) Test Top-1 Error

Figure 1: Comparison of training loss, validation loss, and validation error for different optimizers on CIFAR-100 using ResNet-50. **KOALA++** demonstrates the strongest performance drop at scheduled learning rate decays (epochs 30, 60, and 90), highlighting its superior scheduler responsiveness.

Figure 1 presents the training and validation dynamics on CIFAR-100 with ResNet-50 under a multi-step learning rate schedule at epochs 30, 60, and 90. Compared to SGD, Adam, AdaFisher, and KOALA-M, our proposed **KOALA++** demonstrates consistently lower validation loss on the logarithmic scale, along with improved generalization as shown by the lower Top-1 error.

In particular, the benefits of **KOALA++** are most evident as the scheduled learning rate drops: the optimizer exhibits sharp improvements in both loss and error immediately following the milestones, indicating more effective adaptation to the changing optimization dynamics. This suggests that **KOALA++** better integrates structural information from past gradients and uncertainty estimates, allowing it to respond more effectively to scheduler-induced shifts.

Moreover, **KOALA++** maintains competitive or superior stability throughout all training phases, without exhibiting the overfitting spikes or stagnation observed in other optimizers. These observations collectively validate **KOALA++** as a robust and efficient optimizer for large-scale vision tasks.

**(2) CIFAR-10 + Modern Models + Cosine Annealing + Strong Augmentation** To align with the stronger benchmark protocol proposed by AdaFisher [7], we conduct a second set of experiments that adopt more modern practices in deep learning optimization:

---

[6]`https://github.com/lucidrains/Adan-pytorch`
[7]`https://github.com/jxbz/fromage#voulez-vous-du-fromage`
[8]`https://github.com/juntang-zhuang/Adabelief-Optimizer#`

Table 4: Top-1 accuracy (%) on CIFAR-10 across various architectures. We report the average over 5 seeds. The large number is the mean and the subscript is the standard deviation.

| Optimizer | ResNet50 | ResNet101 | DenseNet121 | MobileNetV3 | Tiny Swin |
|---|---|---|---|---|---|
| SGD | $95.71_{0.1}$ | $95.98_{0.2}$ | $96.09_{0.1}$ | $94.43_{0.2}$ | $82.34_{0.2}$ |
| Adam | $94.45_{0.2}$ | $94.57_{0.1}$ | $94.86_{0.1}$ | $93.32_{0.1}$ | $87.37_{0.6}$ |
| AdaHessian | $95.54_{0.1}$ | $95.29_{0.6}$ | $96.11_{0.1}$ | $92.86_{0.1}$ | $84.15_{0.2}$ |
| K-FAC | $95.66_{0.1}$ | $96.01_{0.1}$ | $96.12_{0.1}$ | $94.34_{0.1}$ | $64.79_{0.5}$ |
| Shampoo | $94.59_{0.1}$ | $94.63_{0.1}$ | $95.66_{0.1}$ | $93.81_{0.2}$ | $63.91_{0.4}$ |
| AdaFisher(W) | $\mathbf{96.34}_{0.2}$ | $\mathbf{96.39}_{0.1}$ | $\mathbf{96.72}_{0.1}$ | $\mathbf{95.28}_{0.1}$ | $\mathbf{88.74}_{0.4}$ |
| **KOALA++** (Ours) | $\mathbf{96.44}_{0.1}$ | $\mathbf{96.66}_{0.1}$ | $\mathbf{96.60}_{0.1}$ | $\mathbf{94.52}_{0.1}$ | $\mathbf{87.64}_{0.2}$ |

- **Advanced models:** ResNet-101, DenseNet-121 [12], MobileNetV3 [11], and the Swin Transformer (Tiny) [18], reflecting a diversity of architectures including convolutional and attention-based designs.

- **Modern learning rate scheduler:** We use CosineAnnealingLR for all optimizers, which has been shown to improve convergence stability and final accuracy in modern setups.

- **Stronger data augmentation:** In addition to standard cropping and flipping, we apply Cutout [5] during training to improve generalization.

- **Larger batch size:** Following current best practices, we increase the batch size to 256, which can benefit optimizers that are sensitive to noisy gradients.

- **More Random Seeds:** Following the AdaFisher benchmark protocol [7], we report top-1 accuracy averaged over five random seeds: 42, 3407, 9331, 7, and 2025. The standard deviation across these runs is shown in the bottom-right corner of each table entry.

As shown in Table 4, **KOALA++** performs on par with AdaFisher across a range of architectures, while exhibiting consistently lower standard deviations, indicating greater stability among random seeds. All **KOALA++** results were obtained without extensive hyperparameter adjustment, highlighting the robustness of the method in the default settings. It is also worth noting that on Swin-Tiny, **KOALA++** reaches strong performance with significantly smaller variance, demonstrating better adaptability to Transformer-based architectures.

**Efficiency profiling.** Beyond accuracy, we also compare the computational efficiency of **KOALA++** with strong baselines. **KOALA++** achieves runtime and memory comparable to Adam(W), while being substantially more efficient than Shampoo and AdaHessian. A detailed comparison, including per-epoch time, FLOPs, and peak memory usage on ResNet-50 and Swin-Tiny, is provided in the Appendix E.3.

## 4.2 Language Model

We evaluated **KOALA++** on the Wikitext-2 dataset, following the same language modeling benchmark setting introduced in the AdaFisher paper [7]. The model is a scaled-down version of GPT-1 with four masked self-attention layers and approximately 28 million parameters. We adopt the same architecture, tokenizer, and 50-epoch training schedule as used in the original benchmark.

In line with the original AdaFisher benchmark (see Github[9]), we report the test perplexity (PPL) corresponding to the best-performing model on the validation set, along with the total training time, to provide a holistic view of performance and efficiency. The results are summarized in Table 5. In addition to reporting test perplexity and total training time, we further provide visualizations to better illustrate the training dynamics (see Appendix E).

## 4.3 Ablation Study

Due to space constraints, we report only the core ablation comparing KOALA-M and two variants of **KOALA++** on CIFAR-100 with ResNet-50. The default **KOALA++** uses a symmetric gain

---

[9]`https://github.com/AtlasAnalyticsLab/AdaFisher/tree/main`

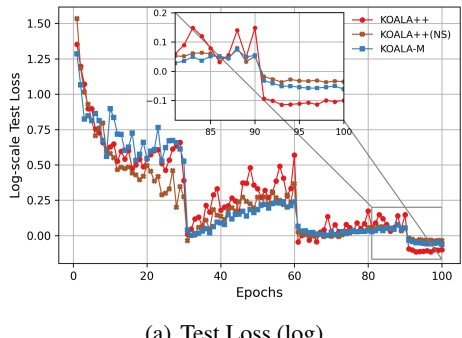
(a) Test Loss (log)

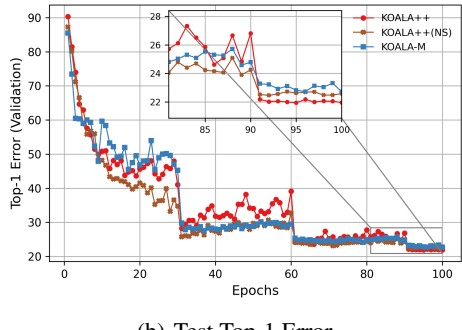
(b) Test Top-1 Error

Figure 2: Ablation study comparing **KOALA++**, its asymmetric variant **KOALA++** (NS), and KOALA-M on CIFAR-100 with ResNet-50. **Left:** Log-scale test loss curves. **Right:** Validation Top-1 error rates.

Table 5: Language modeling performance on Wikitext-2. We report the test perplexity (lower is better) corresponding to the best validation model, as well as the total training time over 50 epochs.

| Optimizer | Test PPL ↓ | Average Training Time per Epoch (s) |
|---|---|---|
| AdamW | 173.98 | 23.11 |
| AdaHessian | 407.69 | 62.91 |
| AdaFisherW | **152.72** | 23.39 |
| **KOALA++** (Ours) | **151.75** | 25.56 |

approximation, while **KOALA++** (NS) adopts an asymmetric covariance. As shown in Figure 2, **KOALA++** achieves faster convergence and better final performance. **KOALA++** (NS) is competitive but less stable in later stages, underscoring the benefit of structural symmetry. Full ablations are provided in the Appendix D.

## 5 Conclusions, Limitations and Future Work

We proposed **KOALA++**, an efficient optimization method that extends the Kalman-based KOALA algorithm by incorporating a lightweight directional covariance approximation. **KOALA++** retains the stability and simplicity of its predecessor, while substantially improving its capacity to capture general structured uncertainty in high-dimensional loss landscapes. Through extensive experiments across vision and language tasks, we demonstrate that **KOALA++** achieves competitive or superior performance compared to both first- and second-order optimizers, while maintaining training efficiency on par with standard first-order methods.

**Limitations.** A notable limitation of our method is that the approximated covariance matrix $P_k$ is not explicitly constrained to be positive semi-definite (PSD). Although we empirically observe that the eigenvalues of $P_k$ remain stable and bounded throughout the training (see the Appendix B.3), the lack of a formal PSD guarantee may pose risks in certain settings or applications involving sensitive numerical stability. Addressing this limitation efficiently is left to future work.

**Future Work.** To address this issue, future work may explore enforcing PSD constraints via projection, spectral regularization, or reparameterization techniques. Additionally, we plan to investigate the integration of **KOALA++** with adaptive gradient clipping and apply it to large-scale pretraining scenarios such as vision-language models, diffusion models, or instruction-tuned LLMs.

## Acknowledgements

Calculations were performed on UBELIX (`https://www.id.unibe.ch/hpc`), the HPC cluster at the University of Bern. Aram Davtyan has been supported by SNSF Grant 10001278.

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

## Appendix Contents

## A  KOALA

In this section, we are going to revisit the original KOALA framework, which was only briefly introduced in the main text. KOALA [4] is a Kalman-inspired optimization algorithm that interprets gradient-based updates as a recursive state estimation process. To ensure scalability in high-dimensional deep learning settings, the original work proposed two tractable variants: KOALA-V and KOALA-M. These variants differ in how they approximate the parameter covariance and in whether they incorporate momentum dynamics. In what follows, we provide a detailed and self-contained exposition of both KOALA-V and KOALA-M, including their derivations, update rules, and practical considerations. This not only offers a clearer understanding of KOALA as a foundational method, but also sets the stage for the improvements introduced in **KOALA++**.

### A.1  Risk Minimization as Loss Adaptivity

The core idea behind KOALA is to cast the optimization of deep neural networks as a state estimation problem under uncertainty. Specifically, neural network training is considered to minimize a noisy risk function $L_k$, which is a stochastic estimate of the empirical risk $\mathcal{L}$. Motivated by the central limit theorem, KOALA assumes that the mini-batch risk $L_k$ can be modeled as a Gaussian random variable with scalar noise

$$L_k(\theta) \simeq \mathcal{L}(\theta) - v_k, \quad \text{where } v_k \sim \mathcal{N}(0, R_k), \tag{21}$$

$\theta$ denotes the model parameters and $R_k$ is the variance of the observation noise.

Rather than optimizing $\min_\theta \mathcal{L}(\theta)$ directly, KOALA reformulates the problem by *adapting* the model parameters so that the stochastic risk $L_k(\theta_k)$ matches a desired target risk $L_k^{\text{target}}$, i.e.,

$$L_k(\theta_k) = L_k^{\text{target}} - v_k. \tag{22}$$

This formulation enables the use of Kalman filtering, which provides a principled way to recursively update the estimate of $\theta_k$ by combining prior knowledge with new and noisy observations. The underlying system can be described as a standard dynamical model:

$$\theta_k = \theta_{k-1} + w_{k-1}, \quad w_k \sim \mathcal{N}(0, Q_k) \tag{23}$$

$$L_k^{\text{target}} = L_k(\theta_k) + v_k, \quad v_k \sim \mathcal{N}(0, R_k). \tag{24}$$

## A.2 KOALA-V (Vanilla)

The idea behind **KOALA-V** is remarkably natural, which is to directly apply the standard Kalman filtering update equations to the system in Equations (23) and (24). However, directly computing the posterior covariance $P_k \in \mathbb{R}^{n \times n}$ is infeasible for high-dimensional models. To address this, KOALA-V introduces the following approximations:

- The posterior covariance $P_k$ is approximated by a scaled identity matrix:

$$P_k \approx \sigma_k^2 I_n. \tag{25}$$

- $H_k^\top H_k$ is approximated by the squared norm of the gradient:

$$H_k^\top H_k \approx \|\nabla_\theta L_k(\theta_k)\|^2 I_n. \tag{26}$$

The resulting parameter update rule is given in Algorithm 2.

This update closely resembles the SGD update:

$$\theta_k = \theta_{k-1} - \eta \nabla \hat{L}_k(\theta_k), \tag{27}$$

but with a key difference: in KOALA-V, the scaling of the gradient is automatically adapted based on the current loss, gradient norm, and covariance estimate. This adaptivity makes KOALA-V more robust to noisy training dynamics than static first-order methods like SGD or Adam.

---

**Algorithm 2** KOALA-V (Vanilla)

---

Initialize $\theta_0$, $P_0$, $Q$ and $R$
**for** $k$ in range$(1, T)$ **do**
   **Predict:**

$$\hat{\theta}_k = \theta_{k-1}; \quad \hat{P}_k = P_{k-1} + Q$$

   **Update:**

$$\theta_k = \hat{\theta}_k - \frac{\hat{P}_k(\hat{L}_k(\hat{\theta}_k) - \hat{L}_k^{\text{target}})}{\hat{P}_k\|\nabla \hat{L}_k(\hat{\theta}_k)\|^2 + R} \cdot \nabla \hat{L}_k(\hat{\theta}_k) \tag{28}$$

$$P_k = \frac{R}{\hat{P}_k\|\nabla \hat{L}_k(\hat{\theta}_k)\|^2 + R} \cdot \hat{P}_k \tag{29}$$

**end for**
**return** $\theta_T$

---

## A.3 KOALA-M (Incorporating Momentum Dynamics)

KOALA-M extends the original KOALA framework by explicitly integrating momentum-based dynamics into the Kalman filtering formulation. Instead of modeling the parameter update as a simple random walk, KOALA-M introduces an auxiliary momentum state $p_k$, resulting in the augmented state dynamics:

$$\theta_k = \theta_{k-1} + p_{k-1} + w_{k-1}, \quad w_{k-1} \sim \mathcal{N}(0, Q), \tag{30}$$

$$p_k = \kappa\, p_{k-1} + u_{k-1}, \quad u_{k-1} \sim \mathcal{N}(0, U), \tag{31}$$

where $\kappa$ controls the momentum persistence. This momentum augmentation enables KOALA-M to better capture and leverage correlations across sequential updates, effectively enhancing stability.

However, KOALA-M inherits computational complexity due to its augmented state. To mitigate this, KOALA-M approximates the posterior covariance with a structured form, typically assuming it to be a Kronecker product of a $2 \times 2$ covariance matrix with an identity matrix $I_n$. Similar to KOALA-V, KOALA-M employs layer-wise covariance updates to obtain a better approximation of parameter uncertainty, partially alleviating the limitation of overly simplistic covariance structures.

In contrast, our proposed **KOALA++** algorithm does not require explicit momentum augmentation or restrictive covariance approximations due to its inherently richer and adaptive covariance representation. **KOALA++** directly maintains structured gradient-covariance products without imposing additional structural assumptions on the covariance matrix $P$, thus naturally capturing both directional information and momentum-like dynamics implicitly.

## B   Mathematical Foundations of KOALA++

In the main paper, we provided both the non-symmetric and symmetric solutions for $P_{k-2}$. In this section, we provide the detailed derivations of those. For notational simplicity, where possible, we drop the subscripts in $k$, and we can restate the optimization problem as follows:

$$\min_P \|P\|_F^2 \quad \text{subject to} \quad HP = v \tag{32}$$

### B.1   Vanilla Optimal Solution of Covariance Matrix P

To solve this constrained optimization problem, we introduce a Lagrange multiplier $\mu \in \mathbb{R}^{1 \times n}$ and construct the corresponding Lagrangian function:

$$\mathcal{L}(P, \mu) = \|P\|_F^2 + \mu(HP - v)^\top. \tag{33}$$

We then find the optimal solution by differentiating the Lagrangian $\mathcal{L}$ with respect to $P$, and set this derivative to zero to find critical points:

$$\frac{\partial \mathcal{L}(P, \mu)}{\partial P} = 2P + H^\top \mu = 0 \quad \Rightarrow \quad P = -\frac{1}{2} H^\top \mu. \tag{34}$$

Now, substituting this optimality condition into the constraint equation $HP = v$, we have:

$$H\left(-\frac{1}{2} H^\top \mu\right) = v \Rightarrow \mu = -2 \frac{v}{HH^\top}. \tag{35}$$

Thus, the optimal solution for $P$ is obtained by substituting back the expression for $\mu$:

$$\boxed{P = \frac{H^\top v}{HH^\top}}. \tag{36}$$

This solution is referred to as the *vanilla optimal solution* (asymmetric) because it does not explicitly impose symmetry constraints on $P$. In the next subsection, we derive a symmetric solution variant that respects the symmetry requirement typically imposed on covariance matrices.

### B.2   Symmetric Optimal Solution of Covariance Matrix P

For derivation of the symmetric solution, we use the same Lagrangian framework, but now explicitly incorporate the symmetry constraint into the formulation. The Lagrangian function, considering the symmetry constraint, remains the same as before (see Equation (33)).

We use the symmetric matrix derivative form from the Matrix Cookbook [23].Taking the derivative with respect to $P$ and setting it to zero yields

$$\frac{\partial \mathcal{L}}{\partial P}(\text{Symmetric}) = \left[\frac{\partial \mathcal{L}}{\partial P}\right] + \left[\frac{\partial \mathcal{L}}{\partial P}\right]^{\top} - \text{diag}\left[\frac{\partial \mathcal{L}}{\partial P}\right] = 0 \tag{37}$$

By plugging in Equation (33), we have

$$2P + \mu^{\top} H + 2P^{\top} + H^{\top} \mu - \text{diag}(2P + \mu^{\top} H) = 0 \tag{38}$$

Since $P$ is symmetric, we can rewrite Equation (38) as:

$$4P + \mu^{\top} H + H^{\top} \mu - \text{diag}(2P + \mu^{\top} H) = 0, \tag{39}$$

which gives the following equations

$$\begin{cases} 4P + \mu^{\top} H + H^{\top} \mu = 0 & \text{(off diagonal)} \\ 2P + H^{\top} \mu = 0 & \text{(on diagonal)}. \end{cases} \tag{40}$$

We observe that the off-diagonal solution satisfies the on-diagonal equation on the diagonal. Therefore, we only need to solve the off-diagonal case. From the off-diagonal constraint $4P + \mu^{\top} H + H^{\top} \mu = 0$, we have $P = -\frac{1}{4}\left(\mu^{\top} H + H^{\top} \mu\right)$. By substituting in $HP = v$, we obtain

$$HP = -\frac{1}{4}(H\mu^{\top} H + HH^{\top} \mu) = v. \tag{41}$$

We can solve $\mu$ as

$$\mu = -4v(H^{\top} H + HH^{\top} I)^{-1}, \tag{42}$$

where $I$ denotes the identity matrix. To compute the inverse, we recall the Woodbury matrix identity [23]:

$$(A + UV)^{-1} = A^{-1} - A^{-1} U (I + V A^{-1} U)^{-1} V A^{-1} \tag{43}$$

By letting $A = HH^{\top} I, U = H^{\top}$, and $V = H$, we find the following expression

$$(HH^{\top} I + H^{\top} H)^{-1} = \frac{1}{HH^{\top}}\left(I - H^{\top} \cdot \frac{1}{2} H \cdot \frac{1}{HH^{\top}}\right). \tag{44}$$

This provides an expression for $\mu$ without matrix inverses, *i.e.*,

$$\mu = -4v\frac{1}{HH^{\top}}\left(I - \frac{1}{2}\frac{H^{\top} H}{HH^{\top}}\right). \tag{45}$$

Finally, we can plug this expression to have a closed-form optimal solution $P^*$, that is,

$$\begin{aligned} P^* &= \frac{1}{HH^{\top}}\left(I - \frac{1}{2}\frac{H^{\top} H}{HH^{\top}}\right) v^{\top} H + H^{\top} v \frac{1}{HH^{\top}}\left(I - \frac{1}{2}\frac{H^{\top} H}{HH^{\top}}\right) \\ &= \frac{v^{\top} H + H^{\top} v}{HH^{\top}} - \frac{Hv^{\top}}{(HH^{\top})^2} H^{\top} H. \end{aligned} \tag{46}$$

### B.3 Eigenvalues Analysis of the Optimal Matrix $P^*$

Recall the closed-form expression for the optimal matrix

$$P^* = \frac{1}{HH^{\top}}(H^{\top} v + v^{\top} H) - \frac{Hv^{\top}}{(HH^{\top})^2} H^{\top} H. \tag{47}$$

To analyze the definiteness of $P^*$, we consider its eigenvalues. Let $w \in \mathbb{R}^{n \times 1}$ be an eigenvector of $P^*$ and $\lambda$ its corresponding eigenvector, then,

$$\lambda w = P^* w. \tag{48}$$

We can immediately see that $w$ lies in the span of $H^{\top}$ and $v^{\top}$, because the range of $P$ lies in their span. Then, we can write

$$w = \alpha H^{\top} + \beta v^{\top}. \tag{49}$$

By substituting it into the eigenvalue equation we obtain

$$\lambda(\alpha H^\top + \beta v^\top) = \frac{1}{HH^\top}(H^\top v + v^\top H)(\alpha H^\top + \beta v^\top) - \frac{Hv^\top}{(HH^\top)^2}H^\top H(\alpha H^\top + \beta v^\top). \quad (50)$$

We now compute both terms. For simplicity, let us name the following three scalar values as

$$x = HH^\top, \quad y = Hv^\top, \quad z = vv^\top. \quad (51)$$

Then, we have

$$P^* w = \frac{1}{x}(H^\top v \alpha H^\top + H^\top v \beta v^\top + v^\top H \alpha H^\top + v^\top H \beta v^\top) \quad (52)$$

$$- \frac{1}{x^2}(Hv^\top H^\top H \alpha H^\top + Hv^\top H^\top H \beta v^\top) \quad (53)$$

$$= \frac{1}{x}(\alpha y H^\top + \beta z H^\top + \alpha x v^\top + \beta y v^\top) - \frac{1}{x^2}\alpha y x H^\top - \frac{1}{x^2}\beta y^2 H^\top \quad (54)$$

$$= \frac{\beta z}{x}H^\top + \alpha v^\top + \frac{\beta y}{x}v^\top - \frac{\beta y^2}{x^2}H^\top \quad (55)$$

$$= \frac{\beta(xz - y^2)}{x^2}H^\top + \frac{\alpha x + \beta y}{x}v^\top. \quad (56)$$

By matching both sides coefficient-wise, we derive a 2D eigenvalue system as follows

$$\alpha\lambda = \frac{\beta(xz - y^2)}{x^2} \quad (57)$$

$$\beta\lambda = \frac{\alpha x + \beta y}{x}. \quad (58)$$

If $\alpha = 0$, then from Equation (57) we know that $\beta = 0$ or $xz - y^2 = 0$. However, $\beta = 0$ implies that $w = \alpha H^\top + \beta v^\top = 0$, which contradicts the assumption that $w$ is a non-zero eigenvector. On the other hand, by the Cauchy-Schwarz inequality, $xz - y^2 = 0$ implies that the two directions $H^\top$ and $v^\top$ become linearly dependent. In this case, the eigenvalues of $P^*$ are 0 and $a$, where $H^\top = av^\top$, due to the linear dependence. The linear dependence is, however, very unlikely in practice, as, in general $H$ and $v$ span a two-dimensional subspace.

Therefore, we very likely have $\alpha \neq 0$. In this case, the eigenvalue $\lambda$ must satisfy

$$\lambda = \frac{\beta}{\alpha} \cdot \frac{xz - y^2}{x^2}. \quad (59)$$

Now, we need to solve the following equation to examine the sign of $\frac{\beta}{\alpha}$

$$(xz - y^2)\left(\frac{\beta}{\alpha}\right)^2 - xy\frac{\beta}{\alpha} - x^2 = 0. \quad (60)$$

We get

$$\frac{\beta}{\alpha} = \frac{xy \pm \sqrt{x^2y^2 + 4(xz - y^2)x^2}}{2(xz - y^2)}. \quad (61)$$

Finally, the eigenvalues of the matrix $P$ are

$$\lambda_{1,2} = \frac{xy \pm \sqrt{x^2y^2 + 4(xz - y^2)x^2}}{2x^2}. \quad (62)$$

Although the matrix $P$ obtained through least-squares minimization does not guarantee positive definiteness, we conducted an empirical analysis by visualizing the evolution of its eigenvalues at different training stages.

Specifically, we performed eigenvalue tracking on the ResNet-50 model trained on CIFAR-100 at 100 epochs. The learning rate was decayed at epochs 30, 60, and 90, which correspond to the early, middle, and late stages of training, respectively.

To better understand the behavior of the matrix $P$, we analyzed the distribution of its eigenvalues across all mini-batches at four critical epochs: 0, 30, 60, and 90. The results are shown in Figure 3.

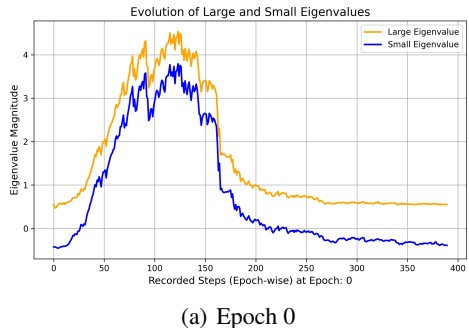
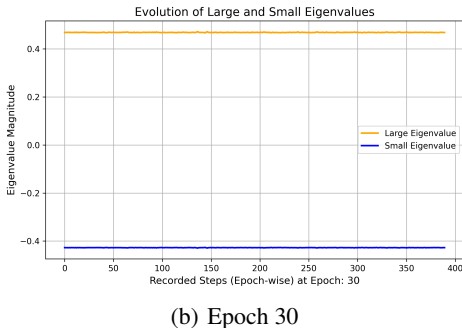

(a) Epoch 0  (b) Epoch 30

Figure 3: Evolution of the large and small eigenvalues of the matrix $P$ at epoch 0 and epoch 30 across all mini-batches.

We found the cases for 60 and 90 epochs to give identical trends as for 30 epochs case. Thus, to avoid unnecessary repetition, we are not showing the plots for these cases.

At epoch 0 (Figure 3, left), the eigenvalues exhibit a wide dynamic range with both large and small eigenvalues increasing significantly over time. The large eigenvalue (orange) and small eigenvalue (blue) rise rapidly and display sharp oscillations, indicating an evolving optimization landscape in the early stage of training.

However, by epoch 30 (Figure 3, right), the eigenvalue trajectories stabilize, and the magnitude of fluctuations reduces dramatically. From epoch 30 onwards, the eigenvalue distributions become increasingly similar. In these later epochs, both the large and small eigenvalues remain relatively flat and nearly constant throughout training, reflecting a stabilized matrix structure and learning dynamics.

## C   Convergence Behavior of KOALA++

### C.1   On the Stability and Implicit Positive Semi-Definiteness of KOALA++

While the current formulation of **KOALA++** does not explicitly enforce the positive semi-definite (PSD) constraint on the covariance-like matrix, we empirically observe that the optimizer remains numerically stable and convergent across diverse architectures and datasets. This suggests that, in practice, the lack of an explicit PSD guarantee does not adversely affect convergence behavior.

This is probably because, when we are calculating the eigenvalues in Equation (62), the quantities $\lambda_{1,2}$ are obtained from the least-squares solution of the auxiliary matrix $P_{k-2}$, which represents the optimal estimate under the minimal-norm fitting objective. In other words, the eigenvalues in Equation (62) correspond to an analytically fitted surrogate of the covariance update, rather than the actual covariance-like matrix used in the algorithmic recursion. In practice, the true update matrix—hereafter referred to as the *ground* truth matrix—is defined by integrating the solution from Equation (12) into (9), then we have:

$$v_k = H_k \left( \frac{H^\top v + v^\top H}{\|H\|^2} - \frac{(Hv^\top)(H^\top H)}{\|H\|^4} + QI - \frac{(v^\top + QH^\top)(v + QH)}{H(v^\top + QH^\top) + R} \right). \tag{63}$$

We call the ground-truth matrix $M_k$ and express it as follows

$$M_k = \frac{H^\top v + v^\top H}{\|H\|^2} - \frac{(Hv^\top)(H^\top H)}{\|H\|^4} + QI - \frac{(v^\top + QH^\top)(v + QH)}{H(v^\top + QH^\top) + R}. \tag{64}$$

Based on this formulation, we can further examine the directional definiteness of the update. Specifically, by evaluating the product $H_k v_k^\top$, we can infer whether the update direction induced by $M_k$ corresponds to a positive-definite step along the current gradient direction. In particular, when

$$H_k v_k^\top = H_k M_k H_k^\top > 0, \tag{65}$$

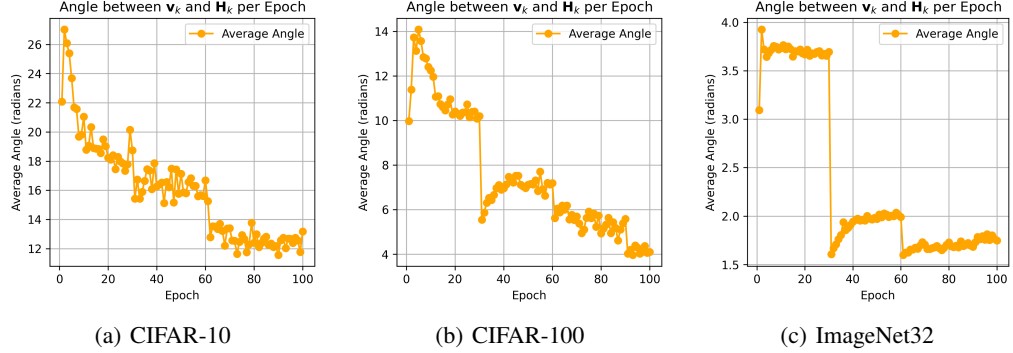

(a) CIFAR-10        (b) CIFAR-100        (c) ImageNet32

Figure 4: Empirical verification of the positive semi-definiteness of $M_k$ in **KOALA++**. The plots show the evolution of the angle between the curvature vector $H_k$ and the update direction $v_k$ across training for three datasets. The consistently acute angles ($< 90°$) indicate that $H_k$ and $v_k$ remain directionally aligned, confirming the effective PSD behavior of $M_k$ in practice.

the update matrix $M_k$ can be considered positive definite with respect to the local curvature defined by $H_k$. This provides a practical criterion for assessing the stability and consistency of the covariance update in **KOALA++**.

Empirically, we verify this property on three benchmark datasets: CIFAR-10, CIFAR-100, and ImageNet32. For each case, we monitor the angle between the curvature vector $H_k$ and the update direction $v_k$ throughout the training. As shown in Figure 4, the angle remains consistently acute ($< 90°$) in all training stages, which implies that $H_k v_k^\top > 0$, thus confirming the numerical positive semi-definiteness of $M_k$ and the empirical stability of the update rule.

However, from a theoretical standpoint, ensuring positive-semidefiniteness could further strengthen the stability guarantees of the method. In particular, one promising direction is to modify the optimization objective so that the covariance estimate minimizes its distance—for example, in the Frobenius norm—to a structured PSD matrix such as a scaled identity. Such a formulation naturally leads to a convex optimization problem with a unique and bounded solution, thereby eliminating ambiguity and improving theoretical soundness.

Although this PSD-enforced variant is beyond the current scope of our study, we view it as an important extension to future work aimed at establishing stronger convergence guarantees for **KOALA++**.

### C.2 Convergence Characteristics of KOALA++ vs. Quasi-Second-Order Optimizers

To better understand how **KOALA++** differs from quasi-second-order optimizers in convergence behavior, we conduct a controlled comparison on CIFAR-10 using a ResNet-50 backbone trained for 100 epochs under a cosine learning rate scheduler.

Although the Shampoo optimizer [8] is a well-established quasi-second-order method, its computational overhead makes direct comparison impractical under identical epoch budgets. To ensure a fair and efficient evaluation, we instead adopt Muon [14], a more recent and scalable curvature-aware optimizer that follows a similar quasi-second-order philosophy but operates with significantly lower wall-clock cost. This substitution allows us to better examine the convergence profile of **KOALA++** against comparable quasi-second-order approaches.

As shown in Figure 5, Muon achieves faster convergence in the early phases, while **KOALA++** initially exhibits larger fluctuations but gradually stabilizes and reaches comparable or better final accuracy. We attribute this difference to the distinct curvature modeling mechanisms of the two optimizers. Muon performs explicit matrix preconditioning with guaranteed positive semi-definite (PSD) curvature estimates, whereas **KOALA++** relies on an implicit least-squares projection derived from scalar observations. Since the resulting covariance-like matrix is not explicitly constrained to be PSD, **KOALA++** may require several epochs to implicitly correct unstable directions before converging to a well-conditioned update regime. This interpretation remains a hypothesis, and a

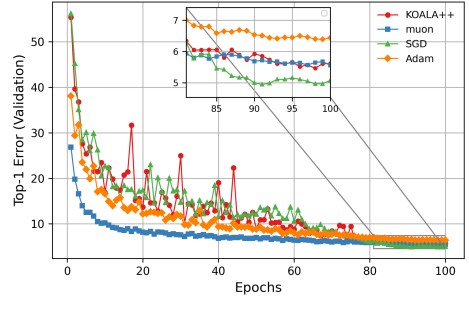

(a) Log-scale Test Loss vs Epochs

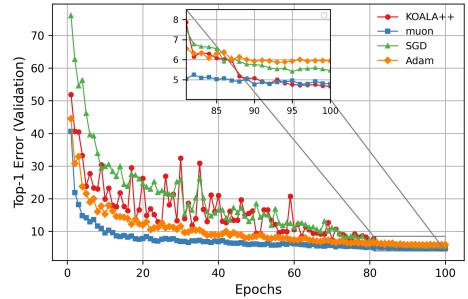

(b) Top-1 Validation Error vs Epochs

Figure 5: CIFAR-10 with ResNet18 (**Left**) and ResNet50 (**Right**) trained for 100 epochs under a cosine learning-rate scheduler. Validation Top-1 errors (the inset highlights the late-epoch region) are reported.

rigorous theoretical analysis of the PSD property and its impact on convergence stability is left to future work.

# D    Ablations

## D.1    Effect of Batch Size on KOALA++ Performance

We evaluated the impact of batch size on the performance of **KOALA++** using ResNet50 on the CIFAR100 dataset. As shown in Table 6, increasing the batch size leads to a noticeable degradation in both Top-1 and Top-5 accuracy. The results suggest that **KOALA++** is more effective in smaller-batch training regimes, possibly due to better gradient variance estimation and more frequent updates.

Table 6: Impact of batch size on **KOALA++** performance (ResNet50, CIFAR100).

| Batch Size | Top-1 Error (%) | Top-5 Error (%) |
|:---:|:---:|:---:|
| 64 | 20.27 | 4.91 |
| 128 | 20.38 | 4.76 |
| 256 | 21.78 | 5.84 |
| 512 | 23.63 | 6.59 |

## D.2    Different Schedulers

In this experiment, we evaluate the impact of different learning rate schedulers on the performance of **KOALA++** using the CIFAR-100 dataset. We consider two representative architectures: ResNet-50 as a typical CNN-based model and Swin-Tiny as a typical ViT-based model. All models are trained for 200 epochs using the same optimizer (**KOALA++**) and hyperparameters.

The partially recorded results are summarized in Table 7.

Table 7: Top-1 and Top-5 error (%) on CIFAR-100 using **KOALA++** with different schedulers.

| Schedulers | ResNet-50 (CNN) | | Swin-Tiny (ViT) | |
|:---|:---:|:---:|:---:|:---:|
| | Top-1 Error | Top-5 Error | Top-1 Error | Top-5 Error |
| Multi-step | 21.80 | 5.89 | – | – |
| CosineAnnealingLR | 20.57 | 5.07 | 35.05 | 12.07 |
| Warmup + Cosine | – | – | 32.18 | 10.64 |

As Table 7 shows, different schedulers can lead to notable differences in final top-1 and top-5 error. We now take a closer look at how these schedulers influence the test loss dynamics during training.

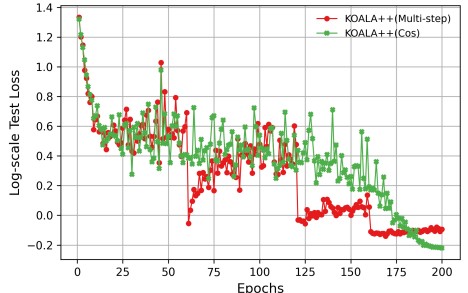
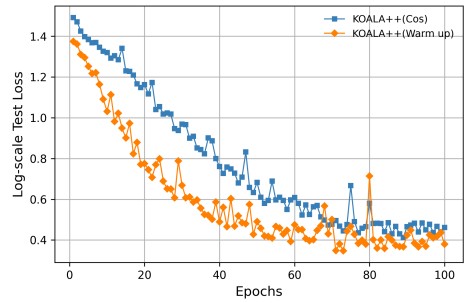

(a) ResNet-50: Multi-step vs. CosineAnnealingLR

(b) Swin-Tiny: CosineAnnealingLR vs. Warmup + Cosine

Figure 6: Log-scale test loss curves of **KOALA++** on CIFAR-100 under different learning rate schedulers. **Left:** CosineAnnealingLR provides a smoother convergence than Multi-step on ResNet-50. **Right:** Warmup + Cosine improves stability and convergence for Swin-Tiny compared to CosineAnnealingLR alone.

Table 8: Effect of varying $\sigma$ on **KOALA++** performance. Dataset: CIFAR-100, Model: ResNet-50, Epochs: 200, $q = 0.2$, lr= 2.0

| $\sigma$ | **Top-1 Error (%)** | **Top-5 Error (%)** |
|------|------|------|
| 0.05 | 22.95 | 6.60 |
| 0.10 | 22.04 | 6.11 |
| 0.20 | **21.80** | **5.89** |

For ResNet-50, we compare the Multi-step scheduler and the CosineAnnealingLR scheduler. As shown in Figure 6, CosineAnnealingLR results in a smoother and more stable descent in test loss, especially during the late training stages. This stable convergence leads to slightly better final performance compared to the traditional multi-step decay.

For the ViT-based Swin-Tiny model, we evaluate both CosineAnnealingLR and a Warmup + Cosine schedule. We find that adding a warm-up stage significantly enhances convergence speed and test loss. This is consistent with the behavior of transformer-based architectures, which are more sensitive to large initial learning rates. The Multi-step scheduler is not included in this setting, as it is generally less effective for transformers.

*Note:* We observed that the Swin-Tiny model tends to overfit in the later stages when trained for 200 epochs. While we report the final Top-1 and Top-5 error at epoch 200 in Table 7 for consistency, the plotted curves in Figure 6 only show the first 100 epochs. This decision was made to focus on the informative convergence behavior and improve the visual clarity of the comparison.

### D.3 Different Initializations of KOALA++

The initialization of the directional covariance product $v_1$ in **KOALA++** depends directly on the initial covariance matrix $P_0$. To study the impact of $\sigma$ on the performance of the model, we conducted an ablation experiment in which we vary $\sigma \in \{0.05, 0.1, 0.2\}$ while keeping other hyperparameters fixed.

We perform this experiment on the CIFAR-100 dataset using the ResNet-50 architecture, training for 200 epochs. The learning rate is set to $2.0$, and the Kalman process noise parameter $q$ is fixed at $0.2$. The results are summarized in Table 8.

We observe that increasing $\sigma$ leads to improved performance, suggesting that initializing $v_1$ with stronger magnitude provides better early-stage directionality, which helps escape suboptimal regions in the optimization landscape. The results are shown in Figure 7.

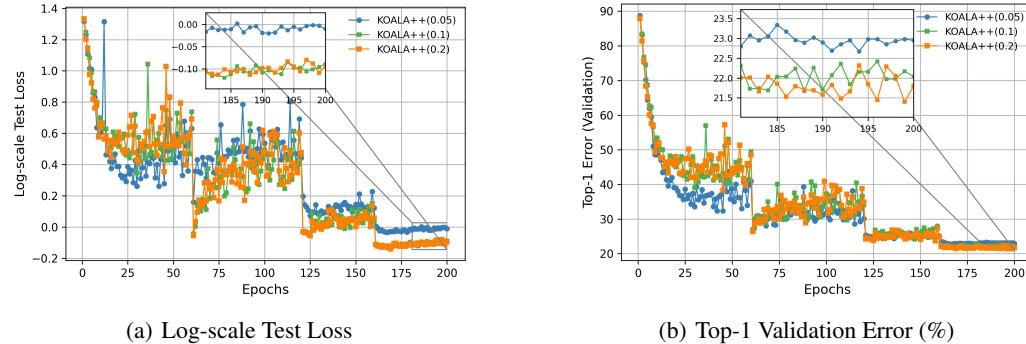

(a) Log-scale Test Loss        (b) Top-1 Validation Error (%)

Figure 7: Effect of varying $\sigma$ on **KOALA++** performance over 200 epochs on CIFAR-100. Larger $\sigma$ values yield more stable convergence and lower final error.

# E    Experiments Details

## E.1    Hardware

All experiments reported in this work were conducted on a server equipped with a single NVIDIA H100 GPU with 80 gigabytes of VRAM and 128 gigabytes of RAM. Unless otherwise stated, all training and evaluation tasks were executed using this configuration.

## E.2    Image Classification

### E.2.1    HP Settings for CIFAR-10/100 and Hyperparameter Sensitivity

We evaluate **KOALA++** on both CIFAR-10 and CIFAR-100 using a variety of architectures, including CNN-based models (e.g., ResNet [9]) and vision transformers (e.g., Swin-Tiny [18]). Across all experiments, we used the same hyperparameter settings for CNN-based models to highlight **KOALA++**'s low-tuning usability, as it achieves competitive or better results than baselines without dataset-specific tuning. When moving to Transformer-based architectures, we conducted a small grid search following the procedure described in the Supplementary Material.

Below we provide further clarification on how the key hyperparameters are selected, consistent with the analysis included in the Supplementary Material to support practical deployment of **KOALA++**:

- **Measurement noise** $R$: estimated online as an exponential moving average of the mini-batch loss variance, using a smoothing factor $\alpha = 0.9$. This removes $R$ from the set of hyperparameters requiring manual tuning.

- **Initial variance** $\sigma_0$: controls only the scale of the initial directional state $v_1 = \sigma_0 H_1$. Ablation studies indicate that **KOALA++**'s convergence is robust to this value, so we fix $\sigma_0$ to match the process noise variance $Q$ for simplicity and symmetry.

- **Process noise** $Q$: the main hyperparameter that requires tuning besides the learning rate. Empirically, we found that $Q \in [0.1, 0.4]$ performs well across datasets.

For CIFAR-10, we initialize both $\sigma_0$ and $Q$ to 0.1, with an initial learning rate of 1.0. For CIFAR-100, which has more classes and a richer data distribution, we adopt slightly larger values $\sigma_0 = Q = 0.2$ and increase the initial learning rate to 2.0. A weight decay of $5 \times 10^{-4}$ is applied to all ResNet and other CNN models.

When applying **KOALA++** to Swin-Tiny, we perform a coarse grid search over the weight decay parameter, evaluating values $\{1 \times 10^{-1}, 1 \times 10^{-2}, 5 \times 10^{-3}, 1 \times 10^{-4}, 5 \times 10^{-4}\}$. Interestingly, **KOALA++** performs best with a smaller weight decay ($1 \times 10^{-4}$), whereas other optimizers typically favor higher decay values (*e.g.*, $1 \times 10^{-2}$). We hypothesize that this stems from **KOALA++**'s implicit regularization through dynamic covariance updates, which already stabilize parameter updates. Thus, combining **KOALA++** with a large explicit regularizer (e.g., strong weight decay) can lead to

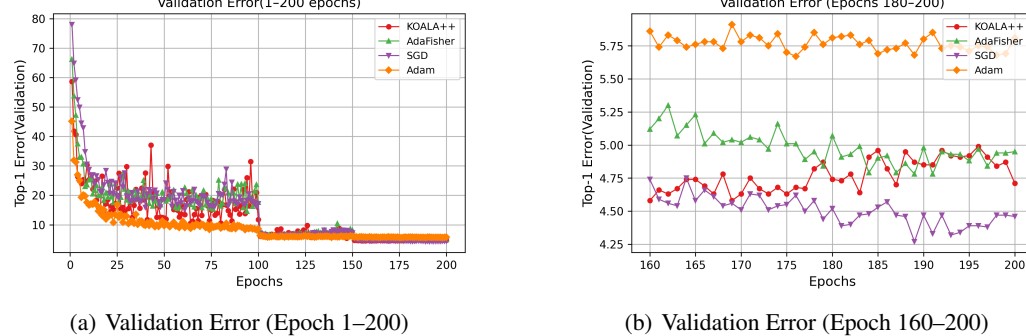

(a) Validation Error (Epoch 1–200)  (b) Validation Error (Epoch 160–200)

Figure 8: Validation Top-1 Error (%) on CIFAR-10 using ResNet-50. **Left:** Full 200-epoch curve. **Right:** Zoomed view of epochs 160–200.

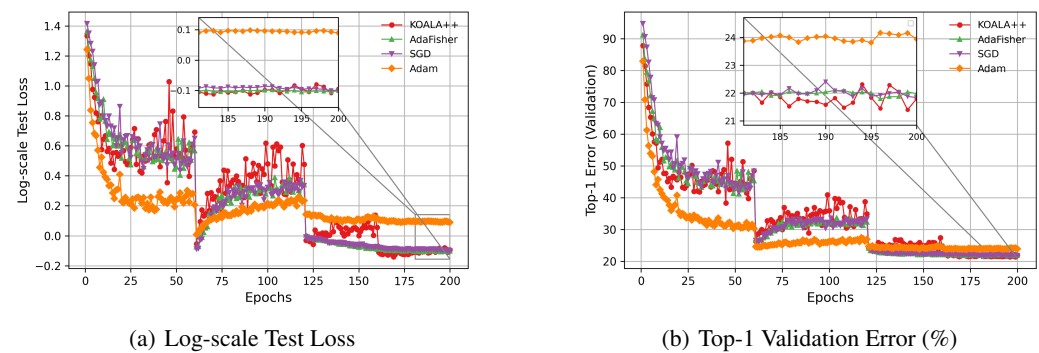

(a) Log-scale Test Loss  (b) Top-1 Validation Error (%)

Figure 9: Results on CIFAR-100 with ResNet-50 over 200 epochs. **Left:** Log-scale test loss. **Right:** Top-1 validation error.

underfitting—especially in transformer-based models—making a smaller weight decay more suitable in such cases.

### E.2.2 More Results for CIFAR10/100

In the main body of the paper, we only visualized 100-epoch results on CIFAR-100 using ResNet-50 to compare **KOALA++** with several baselines. Here, we provide more comprehensive results by extending the training to 200 epochs and including CIFAR-10 as well.

**CIFAR10**    Figure 8 (Left) presents the Top-1 validation error on CIFAR-10 with ResNet-50 across 200 training epochs. To highlight the behavior in the final stage, we also plot a zoomed-in view of the validation error during epochs 160–200 in Figure 8 (Right).

*Note:*This figure shows one random seed; **KOALA++** outperforms SGD on average across three runs.

**CIFAR100**    We additionally report the 200-epoch results on CIFAR-100 using ResNet-50, comparing **KOALA++** with several standard optimizers. The log-scale test loss and Top-1 validation error are shown in Figure 9.

### E.2.3 Experiments for ImageNet32

We train both ResNet18 and ResNet50 models on the ImageNet32 dataset. Following standard augmentation practices for low-resolution image classification, we apply `RandomCrop(32, padding=4)` and `RandomHorizontalFlip()` during training. The input images are subsequently converted to tensors and normalized using the standard ImageNet statistics. For validation, we directly normalize the images without applying any cropping or resizing. All models are trained for 100

Table 9: Final Top-1 and Top-5 validation error (%) on ImageNet32.

| Model | Optimizer | Top-1 Error | Top-5 Error |
|---|---|---|---|
| ResNet18 | SGD | 45.81 | 22.02 |
| | **KOALA++** | **45.71** | **21.94** |
| ResNet50 | SGD | 40.17 | 17.57 |
| | **KOALA++** | **39.03** | **17.12** |

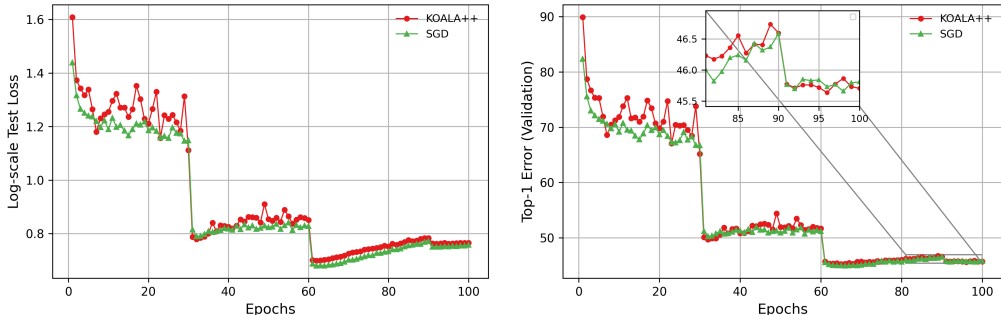

Figure 10: Left: Log-scale test loss; Right: Top-1 validation error for ResNet18 on ImageNet32.

epochs with a learning rate scheduler that decays the learning rate by a factor of 0.1 at epochs 30, 60, and 90, and a weight decay of 0.0001.

*Note:* This setup differs slightly from the data augmentation strategy used in KOALA's ImageNet experiments [4], where validation images are center-cropped prior to evaluation.

Figure 10 shows the log-scale test loss and Top-1 validation error for ResNet18. We observe that SGD reaches a lower test error faster in early training but tends to overfit slightly in the later stages. In contrast, **KOALA++** shows improved generalization in the long run, despite being less stable early on.

Table 9 summarizes the final Top-1 and Top-5 errors for both ResNet18 and ResNet50. The results show that **KOALA++** achieves comparable or better performance to SGD without requiring any task-specific hyperparameter tuning.

### E.3 Efficiency Analysis

We provide additional results on the computational efficiency of **KOALA++**. Besides accuracy, it is important that an optimizer remains efficient in terms of runtime and memory, particularly when scaling to larger models and batch sizes.

**ResNet-50 and Swin-Tiny (batch size 256).** Table 10 reports average training time per epoch, asymptotic FLOPs, and peak memory usage on ResNet-50 and Swin-Tiny. **KOALA++** achieves runtime and memory comparable to Adam(W), while being substantially more efficient than Shampoo and AdaHessian. For K-FAC, we note that the more precise per-iteration complexity is $O(\ell d^2 m)$[10], although it is commonly reported as $O(n^2)$ in the literature.

**ResNet-101 and Swin-Base (batch size 1024).** To further test scalability, we report results on ResNet-101 and Swin Transformer Base with batch size $1024$ in Table 11. **KOALA++** remains competitive with Adam(W), while scaling better than AdaHessian and Shampoo in both runtime and memory. This confirms that **KOALA++** maintains near-first-order efficiency even at larger scales.

---

[10]Here $\ell$ is the number of layers, $d$ the typical hidden dimension, and $m$ the batch size.

[11]AdaHessian uses Hutchinson's trick for Hessian diagonal estimation, which requires additional Hessian-vector products, leading to $O(n^2)$ complexity.

[12]More precisely $O(\ell d^2 m)$, where $\ell$ is the number of layers, $d$ the hidden dimension, and $m$ the batch size.

Table 10: Efficiency profiling on ResNet-50 and Swin-Tiny (batch size 256). We report average training time per epoch, asymptotic FLOPs, and peak memory usage.

| **ResNet-50 (Batch size 256)** | | | |
|---|---|---|---|
| **Optimizer** | **Time/epoch (s)** | **FLOPs (class)** | **Peak Mem (GB)** |
| Adam | 11.08 | $O(n)$ | 10.51 |
| AdaFisher | 11.64 | $O(n)$ | 11.91 |
| AdaHessian[11] | 63.43 | $O(n^2)$ | 22.79 |
| K-FAC[12] | 17.16 | $O(n^2)$ | 12.34 |
| Shampoo | 134.63 | $O(n^{1.5})$ | 10.41 |
| **KOALA++** (ours) | 13.23 | $O(n)$ | 10.27 |
| **Swin-Tiny (Batch size 256)** | | | |
| Adam(W) | 16.35 | $O(n)$ | 3.46 |
| AdaFisher | 12.00 | $O(n)$ | 4.38 |
| AdaHessian | 42.00 | $O(n^2)$ | 5.97 |
| **KOALA++** (ours) | 16.10 | $O(n)$ | 4.50 |

Table 11: Efficiency profiling on ResNet-101 and Swin Transformer Base with batch size 1024.

| **ResNet-101 (Batch size 1024)** | | | |
|---|---|---|---|
| **Optimizer** | **Time/epoch (s)** | **FLOPs (class)** | **Peak Mem (GB)** |
| Adam | 16.79 | $O(n)$ | 48.16 |
| AdaFisher | 16.62 | $O(n)$ | 48.11 |
| K-FAC | 23.79 | $O(n^2)$ | 52.09 |
| Shampoo | 64.29 | $O(n^{1.5})$ | 48.90 |
| **KOALA++** (ours) | 17.61 | $O(n)$ | 48.25 |

| **Swin Transformer Base (Batch size 1024)** | | | |
|---|---|---|---|
| **Optimizer** | **Time/epoch (s)** | **FLOPs (class)** | **Peak Mem (GB)** |
| Adam(W) | 19.07 | $O(n)$ | 23.23 |
| AdaFisherW | 18.95 | $O(n)$ | 22.28 |
| AdaHessian | 57.72 | $O(n^2)$ | 64.08 |
| **KOALA++** (ours) | 19.52 | $O(n)$ | 22.51 |

**Comparison with SGD across batch sizes.** We also compare throughput with SGD on CIFAR-10 using ResNet-50 under varying batch sizes (Table 12). **KOALA++** incurs a small constant overhead per iteration due to $O(n)$ covariance–vector products, but this cost is *independent of batch size*. As the batch size grows, forward/backward cost dominates, and the relative overhead of **KOALA++** shrinks, approaching parity with SGD at batch size 1024.

Table 12: Throughput comparison of **KOALA++** vs SGD across different batch sizes on CIFAR-10 with ResNet-50.

| **Batch Size** | **SGD (s/epoch)** | **KOALA++ (s/epoch)** | **Ratio (KOALA++ / SGD)** | **Overhead** |
|---|---|---|---|---|
| 128 | 11.08 | 17.92 | 1.62× | +61.7% |
| 256 | 10.60 | 13.23 | 1.25× | +24.8% |
| 512 | 9.70 | 11.41 | 1.18× | +17.6% |
| 1024 | 10.56 | 10.60 | 1.00× | +0.4% |

This trend aligns with our design intuition: within each batch, the effective covariance dimension becomes smaller as stochastic noise is averaged out, so the extra operations of **KOALA++** contribute

less to the total runtime. Consequently, **KOALA++** scales gracefully with batch size and approaches the efficiency of SGD in large-batch regimes.

## E.4   Language Modeling

We adopt the experimental benchmark setup provided by AdaFisher [7], available at `https://github.com/AtlasAnalyticsLab/AdaFisher`. For a fair comparison, we use the same experimental pipeline and only test on the WikiText-2 dataset using the small GPT1 model.

To tune the hyperparameters of **KOALA++**, we performed a grid search over the following ranges:

- **Learning Rate (lr)**: {1e-1, 5e-2, 2e-2, 1e-2, 5e-3, 2e-3, 1e-3, 5e-4}
- **sigma and q (set equal)**: {0.01, 0.02, 0.05, 0.1}
- **Weight Decay**: {1e-1, 5e-2, 2e-2, 1e-2, 5e-3, 2e-3, 1e-3, 5e-4, 2e-4, 1e-4}

The optimal configuration we found was: **Learning Rate**: 2e-3, **sigma = q**: 0.1, **Weight Decay**: 1e-4, which achieved the best perplexity among all tested settings. Figure 11 displays the validation loss and test perplexity curve of **KOALA++** on the WikiText-2 dataset.

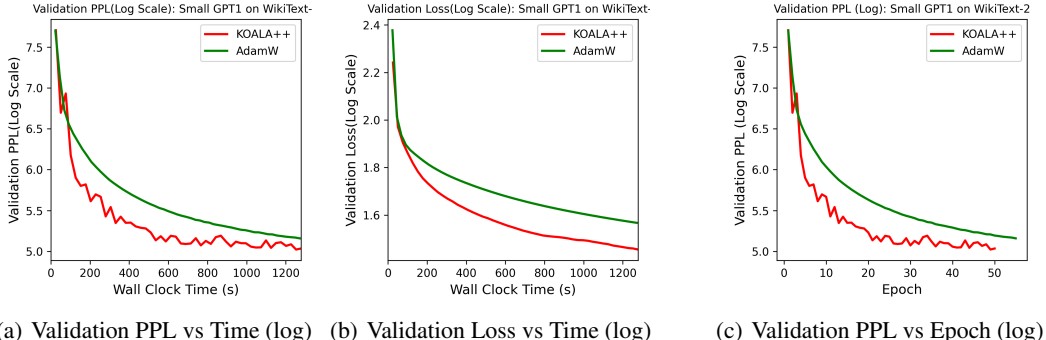

(a) Validation PPL vs Time (log)   (b) Validation Loss vs Time (log)   (c) Validation PPL vs Epoch (log)

Figure 11: Performance of **KOALA++** on WikiText-2 with the small GPT1 model. **Left:** Log-scale validation perplexity over wall clock time. **Right:** Log-scale validation perplexity over training epochs. Following [7], AdamW was trained for 55 epochs, while **KOALA++** were trained for 50 epochs.

