# OpenReview forum: "KOALA++: Efficient Kalman-Based Optimization with Gradient-Covariance Products"
_NeurIPS.cc/2025/Conference — NeurIPS 2025 poster_

### Official Review · Reviewer_oqam · 2025-06-30

**Clarity:** 3
**Significance:** 4
**Originality:** 3
**Rating:** 4
**Confidence:** 3

**Summary:**

This paper introduces a Kalman-based optimization algorithm designed for training neural networks. Rather than relying on expensive second-order gradient calculations or simplistic diagonal covariance assumptions, proposed method explicitly models structured gradient uncertainty through recursive updates of compact gradient-covariance products.

The goals of the work is to achieve the accuracy of second-order methods while maintaining the efficiency of first-order optimizers. Empirical rsults are rpesented on image classification and language modeling tasks.

**Questions:**

- The paper mentions that KOALA++ retains scalability similar to first-order methods. Can the authors provide a more direct comparison of the computational overhead (e.g., FLOPs, memory usage) of KOALA++ versus Adam or SGD for a single training step on a large model?

- Given the acknowledged limitation regarding the positive semi-definite (PSD) nature of key quantities in the optimizer, how much impact does this have on the algorithm's performance in practice, particularly in very deep or sensitive models?

**Ethical Concerns:**

["NO or VERY MINOR ethics concerns only"]

**Limitations:**

N/A (do not think this applies).

**Paper Formatting Concerns:**

Minor typos and grammar issues. Recommend authors to run a grammar check to fix these. No major concerns.

**Quality:**

3

**Strengths And Weaknesses:**

I find this to be a interesting paper solving an important problem in optimization. Some specific points,

Strengths:
- The work addresses an important limitatoin existing in the prior work (KOALA where there are simplistic assumptions made). This is a good improvement.
- The algorithm maintains computational complexity comparable to first-order methods, which is a significant advantage over many traditional second-order optimizers that suffer from high computational costs. This makes the proposed method appealing to practitioners in my opinion. Second order methods are back in fashion more recently.
- The paper produces strong empirical results across diverse tasks and architectures which I greatly appreciate.
- Detailed comparison study across various other key optimizers is provided (e.g. Table 4).
- Motivation and writing in the paper is clear in my opinion.

Weaknesses:
- My main concern is the lack of direct comparison with all major second-order optimizers: While it compares against AdaFisher, the more recent popular ones like K-FAC, Shampoo are only briefly mentioned in tables without extensive discussion of their performance characteristics. Probably other reviewers noted this too. Comparison of training time, efficiency and convergence with these more popular methods will further strenghten the paper.

---

> ### Author Rebuttal · Authors · 2025-07-31
>
> We sincerely thank the reviewer for the thoughtful and constructive feedback. We are especially grateful for your recognition of our key contributions and your clear understanding of the motivation and design behind KOALA++. Your positive comments, particularly acknowledging KOALA++ as a valuable improvement over the original KOALA, its empirical robustness across diverse settings, and the clarity of our writing, are very encouraging to us.
>
> Below, we address the specific concerns raised:
>
> ---
> #### **FLOPs and Memory Usage Comparison**
> We thank the reviewer for this insightful question. To address it comprehensively, below we have included an **efficiency profile** summarizing the **optimizer-side FLOPs, per-epoch wall-clock time, and peak memory usage** across larger architectures: ResNet‑101 (CNN), Swin Transformer Base (ViT) with larger batch size(1024). For the results of GPT-1, see Table 5 of the main paper. All experiments use the same dataloader, hardware, and exclude warm-up steps.
>
> Here are our observations:
>
> - KOALA++ maintains **$O(N)$** optimizer-side complexity via structured vector operations and avoids matrix inversions entirely. This contrasts with methods like Shampoo that involve more matrix operations (e.g., Cholesky, blockwise inverses).
>
>
>
>
> - KOALA++ is consistently **close to AdamW** in epoch time and **significantly faster** than AdaHessian/Shampoo. As batch size grows, KOALA++ approaches **near parity** with AdamW due to forward/backward FLOPs dominating total runtime.
>
>
>
>
> #### **Efficiency Profile**
>
>
> ##### **ResNet‑101 (Batch Size = 1024)**
>
> | Optimizer         | Time per epoch (s) | FLOPs (class) | Peak Mem (GB) |
> |------------------:|-----------------:|:--------------|--------------:|
> | Adam              | 16.79            | O(N)          | 48.16         |
> | AdaFisher         | 16.62            | O(N)          | 48.11         |
> | K‑FAC             | 23.79            |       | 52.09         |
> | Shampoo           | 64.29          | O(N^1.5)      | 48.9         |
> | **KOALA++ (ours)**| **17.61**        | **O(N)**      | **48.25**     |
>
>
>
> ##### **Swin Transformer Base(Batch Size = 1024)**
>
> | Optimizer         | Time per epoch (s) | FLOPs (class) | Peak Mem (GB) |
> |------------------:|-----------------:|:--------------|--------------:|
> | AdamW           | 19.07            | O(N)          | 23.23          |
> | AdaFisherW         | 18.95            | O(N)          | 22.28          |
> | AdaHessian        | 57.72            |           | 64.08          |
> | **KOALA++ (ours)**| **19.52**        | **O(N)**      | **22.51**      |
>
>
>
> ---
>
> #### **PSD Impact**
> We thank the reviewer for this insightful question regarding the PSD (positive semi-definite) property of the covariance-like quantities in KOALA++.
>
>
> As mentioned in the Limitations section, we point out that the approximated $P_{k - 2}$ is not necessarily guaranteed to be PSD. To better understand the impact of this missing constraint, we carry out analysis and experiments in the Supplementary Material in section B.3. This investigation shows that the algorithm is quite stable in terms of the $P_{k-2}$ matrix eigenvalues despite lacking the PSD property. This initial analysis is further confirmed by the fact that in **all** our empirical results across various datasets and model architectures we observe that this **does not degrade convergence** or stability of the optimizer.
>
>
>
> ---
>
> #### **Minor typos and grammar issue**
>
> We thank the reviewer for pointing out the minor typos and grammar issues. We will carefully revise the manuscript to correct all such issues.

---

> > ### Comment · Reviewer_oqam · 2025-08-06
> > **Follow up after rebuttal**
> >
> > Thanks to the authors for answering my questions and clarifying my concerns especially by providing more precise system level metrics (FLOPS, memory) - these are encouraging.
> >
> > Regarding my comment on PSD properties of KOALA++ formulation, I did look at the mentioned supplementary material - thanks for the work there - I still feel this is an area where the paper could improve upon and, therefore as such I feel that I will stick to my original "positive" score.

---

> > > ### Author Response · Authors · 2025-08-08
> > > **Thanks for the discussion**
> > >
> > > Thank you very much for your thoughtful follow-up and for taking the time to engage further with our work. We're glad that the additional system-level metrics (FLOPS, memory) helped clarify your concerns.
> > >
> > > We appreciate your point regarding the PSD formulation, and we agree this is an area where the presentation and discussion can be improved. We will revise this part in the final version to better clarify the underlying structure and limitations.

---

### Official Review · Reviewer_RxbJ · 2025-07-02

**Clarity:** 2
**Significance:** 2
**Originality:** 2
**Rating:** 3
**Confidence:** 4

**Summary:**

The authors propose a modification of the Koala algorithm that considers a different posterior covariance update to the Kalman filter based on a directional covariance projection.

**Questions:**

Koala++, being a quasi-second-order optimiser, I would've expected the test loss to be much lower than first-order optimisers because they can navigate the loss landscape much more efficiently than their first-order counterparts. However, Figure 1 shows the exact opposite: it is only at the very end of training that we see Koala++ obtain a lower test loss and error. I find this to be counterintuitive. Could you explain why we see this behaviour, and is this the case for all other experiments?

(typo?): I believe all $QI$ in Table 1 should be only $Q$ as you treat $Q$ as a matrix and not a scalar.

**Ethical Concerns:**

["NO or VERY MINOR ethics concerns only"]

**Final Justification:**

I appreciate the authors’ follow-up, but my key concerns remain unaddressed. The central issue was the lack of empirical or theoretical support for the claim that KOALA++’s curvature-aware component only provides benefits late in training. While the authors now attribute differences in convergence profiles to entanglement with learning rate schedulers, this remains a hypothesis without direct experimental validation or comparison to methods like Muon and Shampoo, as I originally requested. Their explanation still does not reconcile the documented early-stage advantages of these quasi–second-order optimizers in the literature. In addition, the paper provides no substantive theoretical analysis of KOALA++’s capabilities, relying instead on linear algebraic formulation without deeper insight, and it oversells the significance of the simple least-squares step, which is a rather basic component. Without (1) a principled theoretical grounding and (2) empirical evidence demonstrating a genuine difference in early-phase behavior, the core claims remain speculative. Consequently, I do not find sufficient basis to raise my score.

**Limitations:**

See above.

**Paper Formatting Concerns:**

No issues.

**Quality:**

2

**Strengths And Weaknesses:**

- **Strengths**: The use of a KF-inspired optimisation algorithm to tackle the learning of a high-dimensional model is commendable.  The empirical performance looks good.

- **Weaknesses**: The paper feels like a quick extension of the original Koala paper. The introduction of the directional covariance projection and its optimisation objective is simply introduced without any major motivation. Various approximations are made, but no error analysis is presented as to the choice of approximate covariance. Furthermore, some of the results seem counterintuitive to what one would expect in using proxy second-order optimisation methods as opposed to first-order optimisation methods (see question below).

The paper is also missing references that have also tackled the problem of learning with a KF-type optimisation algorithm that *do not* assume a diagonal covariance matrix. I believe these would've been better choices of benchmarks, as the proposed Koala++ method is not a first-order optimisation algorithm.

For example:
1.⁠ ⁠Chang, Peter G., et al. "Low-rank extended Kalman filtering for online learning of neural networks from streaming data." Conference on Lifelong Learning Agents. PMLR, 2023.
2.⁠ ⁠Mishkin, Aaron, et al. "Slang: Fast structured covariance approximations for bayesian deep learning with natural gradient." Advances in neural information processing systems 31 (2018).
3.⁠ ⁠Duran-Martin, Gerardo, Aleyna Kara, and Kevin Murphy. "Efficient online bayesian inference for neural bandits." International conference on artificial intelligence and statistics. PMLR, 2022.

A more thorough literature review, comparison to prior work, and the points mentioned above **must** be addressed by the authors.

---

> ### Author Rebuttal · Authors · 2025-07-31
>
> We sincerely thank the reviewer for their time and thoughtful evaluation of our work. We greatly appreciate the reviewer that they notice the value of using Kalman Filter as a deep neural network optimizer.
>
> Below, we address the specific concerns raised:
>
> ---
> #### **Thinking of KOALA++**
> We appreciate the reviewer’s summary and understand the impression that KOALA++ may appear as a quick extension of the original KOALA method. However, we respectfully clarify that while KOALA++ is indeed based on KOALA, we do **not** view it as a mere incremental extension. Instead, KOALA++ addresses core limitations in KOALA’s original design and introduces a substantially different treatment of uncertainty modeling in optimization.
>
> As noted in our paper, KOALA treats neural network optimization as estimating the state in the Kalman Filter's dynamic system. However, it makes several simplifying assumptions that we believe limit its expressiveness（see lines 77-79 in the main paper).
>
> 1. **Covariance Approximation**:
>    Although the covariance $P_{k-1}$ is formally a matrix, KOALA approximates it as a **scaled identity** to reduce computational complexity. We found this to be limiting, especially in high-dimensional neural network optimization settings.
>
> 2. **Gradient Product Approximation**:
>    KOALA further approximates the rank-1 matrix $H_k^\top H_k$ as another scaled identity:
>    $$
>    H_k^\top H_k \approx \|H_k\|^2 \cdot I
>    $$
>    This approximation discards the interactions between different weights of the network.
>
> To address these issues, **KOALA++** introduces a core methodological shift: instead of using scaled identity matrices, we do not make any structural assumptions on $P_k$ and only update it implicitly by tracking **vector–matrix products**. While expressions like $H_k P_{k-2}$ still appear in the update, we design the algorithm to **never explicitly store or construct** the covariance $P_{k-2}$ by using the relationship, $H_{k-1} P_{k-2} = v_{k-1}$ and recovering $P_{k-2}$ via a simple least-squares fit, which preserves the efficiency of the original design while offering significantly more expressive updates.
>
>
> ---
> #### **Motivation of directional covariance optimization objective**
>
> We appreciate the request for a clearer motivation. Below we explain why we project the covariance onto a directional subspace and why we solve a **Frobenius minimum‑norm** problem to approximate it.
>
> Our decision to minimize the Frobenius norm of $P_{k-2}$ was guided by two main aspects: (1) it yields a convex optimization problem with a unique solution, ensuring an unambiguous covariance estimate; and (2) it helps keep the estimate bounded, which is important for maintaining numerical stability in our method. We will revise the paper to make this design decision and its motivation clearer in the main text, and we thank the reviewer again for bringing this to our attention.
>
> ---
>
> #### **Error Analysis**
>
> We thank the reviewer for raising this point. In this work, our primary focus has been on the empirical evaluation of the method. Indeed, we plan to work on a formal theoretical analysis of the approximation and other aspects of the method in future work.
>
> ---
>
> #### **Related Work**
>
> We thank the reviewer for pointing to the related work in Kalman-inspired optimization methods with structured (non-diagonal) covariance approximations. We acknowledge the relevance of these methods and will include a discussion of them in our revised manuscript.
>
> While KOALA++ is indeed inspired by Kalman filtering principles and incorporates uncertainty modeling, it differs from the cited works in design choices and application settings:
>
> - **Chang et al. (2023)** relies on EKF-like updates with diagonal plus low-rank covariance approximations to manage computational cost. **Mishkin et al. (2018)** also propose a diagonal-plus-low-rank approxmation of the inverse covariance for efficient posterior approximation in the variational Bayesian inference. In contrast to these methods, KOALA++ does not make explicit assumptions on the structure of the covariance and only tracks its directional projections. Moreover, **Chang et al. (2023)** formulates the training as online Bayesian inference, while KOALA++ stays within classical deep learning optimization settings. Nonetheless, we agree that the underlying methodology is related and will discuss this in the revised paper.
>
> - **Duran-Martin et al. (2022)** reduces the dimensionality of the state space and performs inference in a lower-dimensional subspace using full EKF updates, thereby avoiding structural assumptions on the covariance in the original weight space. In contrast, KOALA++ operates directly in the original parameter space, benefiting from flexibile navigation in the high-dimensional space, while maintaining linear computational complexity without imposing explicit structural constraints on the covariance.
>
>
>
> ---
> #### **Counterintuition about the training dynamics**
>
> We would like to thank the reviewer for the insightful comment. We recognize that the most significant improvements with KOALA++ come closer to the end of the training where the learning rate is small and the method navigates the loss landscape at the smallest scale. Our reasoning is that during the early stages of training, the optimization landscape is typically more non-convex and noisy. The aggressive updates from first-order methods are particularly effective at these stages. As training advances and the loss surface becomes smoother at the smaller scale, leveraging the curvature of the loss, as (quasi-)second-order methods do (e.g. KOALA++ modelling the uncertainty of the weights' updates), becomes more beneficial and enhances generalization. We would like to thank the reviewer again for pointing our attention to this interesting behavior and leave its further investigation for the future work.
>
> ---
>
>
> #### Potential typo in $QI$
>
> While $Q$ is a matrix in the standard EKF, in KOALA++, we adopt the design choice from KOALA and use a scaled identity matrix $QI$.
>
> ---

---

> ### Comment · Reviewer_RxbJ · 2025-08-06
> **RE: Rebuttal by Authors**
>
> Thanks for your response. I am not totally convinced about some of your justifications. First, while you highlight several aspects that are novel to KOALA++, there is no actual analysis (only linear algebra) whereas it would be valuable to include some insight or theoretical understanding of the optimizer itself. Second, the “simple least squares fit” appears to be a rather basic component, but I am willing to acknowledge it as a practical advantage of the proposed approach.
>
> However, a point that concerns me is your explanation that KOALA++ only gains from its directional covariance projections late in training conflicts. This seems to be contrary to what has been the reported behavior of other quasi–second-order optimizers in the literature:
>
> 1. Figure 2 in [1] shows that Muon (despite using a similar curvature-aware update) outperforms Adam from the very first step, reaching a validation loss of $\approx 3.95$ after only 2s, compared to Adam’s higher loss at the same wall-clock time. If curvature information were only useful once the learning rate is small, Muon would not hold such an advantage immediately.
>
> 2. Figure 4 in [2] illustrates Shampoo’s log-perplexity on LM1B, where it already beats Adam at 50k steps and maintains that gap through 500k steps. A preconditioned method that only yielded late-stage improvements would not show a steady, early reduction in perplexity like Shampoo does.
>
> These two cases directly contradict the claim that “aggressive first-order updates dominate initial training and quasi-second-order gains emerge only at the end.” To reconcile this, please either:
>
> * Provide a clear theoretical argument for why KOALA++’s directional least-squares projection fails to leverage curvature information during the early phase, or
> * Include Muon and Shampoo in your empirical comparisons to demonstrate that KOALA++ truly differs in its convergence profile.
>
> Absent such an explanation or data, the hypothesis about delayed curvature benefits is unsupported. I appreciate that the time to include new experiments at this point is limited, but without proof of this point, I find it hard to raise my score. I would appreciate at least a convincing justification.
>
> References:
>
> [1] https://kellerjordan.github.io/posts/muon/
>
> [2] Gupta, Vineet, Tomer Koren, and Yoram Singer. "Shampoo: Preconditioned stochastic tensor optimization." International Conference on Machine Learning. PMLR, 2018.

---

> > ### Author Response · Authors · 2025-08-08
> > **Thanks for the reviewer's discussion**
> >
> > Thanks again for engaging in the discussion.
> >
> > Regarding the comments on the convergence profile, we are happy to add the convergence profiles of Muon and Shampoo to the empirical comparisons. However, the rebuttal interface does not allow us to upload pdf or image files. Moreover, we don't have enough time to run such experiments right now. One explanation for the different convergence profiles is that Shampoo and Muon use specific learning rate schedulers that enhance early convergence. However, establishing whether that is the case, requires further investigation. We looked at the implementation details of the original Shampoo paper, but it was not clear which learning rate scheduler they employed. Similarly, in Muon it is unclear what learning rate schedulers have been employed. It seems that in several cases, Muon and Shampoo use the cosine annealing learning rate scheduler. For example, as reported in our Table 4, the numbers on Shampoo are based on the settings from [1],  where the cosine annealing learning rate scheduler is used. Similarly, in [2], Muon uses a cosine annealing learning rate scheduler with warm-up to train an LLM. For compatibility, in Table 4, we also used a cosine annealing learning rate scheduler for KOALA++ (and achieved a significant improvement over Shampoo). Moreover, we have looked at the convergence profiles on CIFAR100 with cosine annealing and we observed that KOALA++ has an initial faster convergence rate than SGD, which we have not observed when using a multi-step learning rate scheduler. Our current explanation for KOALA++'s convergence (and for the other methods) is that it is entangled with the learning rate scheduler. Thus, our above explanation for the convergence in Figure 1 and 2, where we used the multi-step learning rate scheduler, may still hold. We agree that it would be interesting to see the convergence profiles of these methods, and we will add them in the final version of the paper.
> >
> >
> > [1] D. Gomes et al. AdaFisher: Adaptive Second Order Optimization via Fisher Information, ICLR 2025
> >
> > [2] J. Liu el al. Muon is Scalable for LLM Training, arxiv 2025

---

### Official Review · Reviewer_esMW · 2025-07-03

**Clarity:** 3
**Significance:** 3
**Originality:** 3
**Rating:** 4
**Confidence:** 4

**Summary:**

This paper introduces KOALA++, an extension of the Kalman-inspired optimizer KOALA, designed to improve efficiency and robustness in deep neural network training. KOALA++ innovatively tracks a low-rank surrogate of the gradient-covariance product to approximate directional uncertainties in optimization without storing or inverting full covariance matrices. It bridges the gap between first-order optimizers (e.g., SGD, Adam) and second-order optimizers (e.g., K-FAC, Shampoo) by maintaining efficiency while capturing richer curvature information. Extensive experiments on image classification (CIFAR-10/100, ImageNet) and language modeling (WikiText-2) show that KOALA++ performs competitively with state-of-the-art optimizers in terms of both accuracy and convergence stability, with manageable computational overhead.

**Questions:**

Could the authors provide quantitative measurements (e.g., time per epoch, memory footprint) comparing KOALA++ to other second-order optimizers like AdaFisher, Shampoo and NG+ on larger models like Swin Transformers or GPT-based architectures?

In practice, how much wall-clock speedup is retained relative to plain SGD?

Have the authors considered hybrid methods that might store low-rank factorizations (e.g., via sketching techniques) instead of purely vector products?

**Ethical Concerns:**

["NO or VERY MINOR ethics concerns only"]

**Limitations:**

yes

**Quality:**

3

**Strengths And Weaknesses:**

Strengths:
The paper is well-written, mathematically solid, and offers promising empirical evidence.

The paper provides a significant methodological advance by designing a Kalman-inspired optimizer that efficiently models structured uncertainty without requiring full covariance matrices or expensive matrix inversions.

Derivations are carefully detailed, with two variants of covariance estimation derived rigorously.

Weaknesses:
While the method is claimed to be efficient, the paper would benefit from more detailed profiling of compute time or FLOP comparisons versus competitors like AdaFisher or K-FAC, especially on large-scale models.

The method introduces several hyperparameters (e.g., Q, R, initial σ²), but guidance on selecting these in practice is limited, and sensitivity analyses are sparse.

The language modeling results are encouraging but are reported on a relatively small dataset (WikiText-2). Testing KOALA++ on larger LLM pretraining settings would solidify claims of scalability.

---

> ### Author Rebuttal · Authors · 2025-07-31
>
> We sincerely thank the reviewer for their time and thoughtful evaluation of our work. We greatly appreciate the reviewer’s positive feedback on our theoretical deriviation of KOALA++, and we are pleased that they found it solid and rigorous.
>
> Below, we address the specific concerns raised:
>
> ---
>
> #### **Efficiency profiling**
>
>
> To illustrate the compotational efficiency of KOALA++, we reported the average training time per epoch for the NLP experiment on WikiText-2 (see Table 5). As our method's computational complexity is task-independent, we considered this signle example to be sufficiently representative. Nonetheless, as requested by the reviewer, we have run additional experiments across different tasks/architectures to better represent the efficiency profile of KOALA++. We provide the results here below:
>
> ##### **ResNet‑50 (Batch Size = 256)**
>
> | Optimizer         | Time per epoch (s) | FLOPs (class) | Peak Mem (GB) |
> |------------------:|-----------------:|:--------------|--------------:|
> | Adam              | 11.08            | O(N)          | 10.51         |
> | AdaFisher         | 11.64            | O(N)          | 11.91         |
> | AdaHessian        | 63.43            |           | 22.79         |
> | K‑FAC             | 17.16            |       | 12.34         |
> | Shampoo           | 134.63           | O(N^1.5)      | 10.41         |
> | **KOALA++ (ours)**| **13.23**        | **O(N)**      | **10.27**     |
>
>
>
> ##### **Swin Transformer (Batch Size = 256)**
>
> | Optimizer         | Time per epoch (s) | FLOPs (class) | Peak Mem (GB) |
> |------------------:|-----------------:|:--------------|--------------:|
> | Adam(W)           | 16.35            | O(N)          | 3.46          |
> | AdaFisher         | 12.00            | O(N)          | 4.38          |
> | AdaHessian        | 42.00            |           | 5.97          |
> | **KOALA++ (ours)**| **16.10**        | **O(N)**      | **4.50**      |
>
>
> As the time for the rebuttal is limited, we were not able to run the experiments with larger models. However, we will include them in the revised paper.
>
> ---
>
> #### **Hyperparameter guidance & sensitivity**
>
> Notice that in the experiments on CIFAR‑10/100 (Tables 2, 3 and 4, CNN-based models), we used the same hyperparameter settings across different models. The results demonstrate KOALA++’s low‑tuning usability (it achieves better or on par results with the baselines), which we consider the practical strength of our method. When moving to the Transformer-based architectures, we adopted a small grid search (see Supplementary Material lines 195-198, 220-222). We will better clarify our evaluation scheme in the revised paper.
>
> Below we provide some insights on choosing the hyperparameters of KOALA++, which we will also include in the revised manuscript to better support practical deployment of our method:
>
> - As for the measurement noise $R$, we estimate it online as an exponential moving average of the mini-batch loss variance, using a smoothing factor $\alpha = 0.9$ as implemented in our code. This procedure effectively removes $R$ from the set of hyperparameters that require tuning.
> - $\sigma_0$ determines only the scale of the initial directional state $v_1 = \sigma_0 H_1$. We have done some ablation studies for this parameter (see Supplementary Material Table 3), and we found that the convergence of KOALA++ is robust to the choice of $\sigma_0$, so for simplicity and symmetry, we suggest to set the prior variance $\sigma_{0}$ and the variance of the process noise $Q$ to the same value.
> - Given above choices, $Q$ is the only hyperparameter except for the learning rate that requires some careful tuning. Empirically we found that $Q \in [0.1, 0.4]$ works relatively well across different datasets.
>
> ---
>
>
> #### **Speed vs SGD**
>
> Below we provide the throughput comparisons vs SGD on CIFAR-10 with ResNet50 using different batch sizes. All the runs are done on a single H100 GPU.
>
> | Batch Size | SGD (s/epoch) | KOALA++ (s/epoch) | KOALA++ / SGD | Overhead     |
> |-----------:|--------------:|------------------:|---------------:|--------------:|
> | 128        | 11.08         | 17.92             | **1.62×**      | **+61.7%**    |
> | 256        | 10.60         | 13.23             | **1.25×**      | **+24.8%**    |
> | 512        | 9.70          | 11.41             | **1.18×**      | **+17.6%**    |
> | 1024       | 10.56         | 10.60             | **1.00×**      | **+0.4%**     |
>
>
> The trend matches our design intuition: KOALA++ performs a small set of additional $O(N)$ operations per iteration compared to plain SGD. These optimizer costs are **independent of the batch size**, while forward/backward cost grows with the batch size. As a result, the relative overhead shrinks from 1.62× (bs=128) to 1.00× (bs=1024), approaching parity with SGD.
>
> Additionally, while this already demonstrates the efficiency of the method, the throughput of KOALA++ can be further improved by reusing repeated dot products across consecutive iterations. We plan to include updated results reflecting these improvements in the revision.
>
> ---
>
> #### **Scalability in LM**
>
> We thank the reviewer for the valuable suggestion. We agree that demonstrating the effectiveness of KOALA++ in larger-scale training would further strengthen the paper. However, due to the limited scope and time constraints of the rebuttal period, we are unable to run such experiments at this stage. That said, we plan to evaluate KOALA++ on larger architectures from the GPT family in future work.
>
> ---
> #### **Consideration of hybrid methods**
>
> We thank the reviewer for this insightful suggestion.
>
> We did consider hybrid approximations of the covariance matrix, such as restricting its structure to a diagonal plus rank-1 form (i.e., $P_k = \sigma I + v_k^\top v_k$), with corresponding updates to the vector $v_k$. While this approach captures slightly richer curvature information, it has been explored in prior work (e.g., Lo-Fi by Chang et al., 2023) and is closely related to KOALA’s original scaled identity formulation, which potentially limits its novelty.
>
> In contrast, KOALA++ takes a fundamentally different approach by operating directly on vector–matrix products, **without ever explicitly constructing** $P_k$. This avoids imposing structural assumptions and opens up greater flexibility.  However, we acknowledge that there may be more effective hybrid approximations or sketching-based factorizations that could offer better trade-offs. We appreciate the reviewer for pointing out this direction and will reflect on it more carefully in future work.
>
> ---

---

> ### Author Response · Authors · 2025-08-08
> **Looking Forward to Your Feedback on Our Rebuttal**
>
> We appreciate your initial review, and we are looking forward to your feedback on our rebuttal. We would greatly appreciate the opportunity to engage in further discussion to clarify any remaining concerns or questions you may have.
>
> Thank you again for your time and for reviewing our work.

---

### Official Review · Reviewer_csNz · 2025-07-03

**Clarity:** 3
**Significance:** 3
**Originality:** 3
**Rating:** 4
**Confidence:** 4

**Summary:**

The paper introduces a new kalman filter based gradient descent methods that extends a current SOTA method called KOALA. KOALA++ i using a low-rank (rank = 1) estimate of the covariance matrix in the Kalman filter as opposed to a diagonal estimate in the original KOALA paper. In order to obtain the covariance update of previous time steps, the authors formulate a constraint optimization problem based on the minimum frobenius norm and symmetry constraints. The authors show the competitiveness of their approach on several image-based (CIFAR-10 and CIFAR 100) and language based tasks (training a GPT-1 model).

**Questions:**

- Why is the Frobenius norm a good measure for the optimization problem? Minimizing the Frobernius norm without constraints would result in 0 covariance. Would an objective that would measure the distance to a prior covariance (> 0) not make more sense?
- Its unclear to me wether the hyperparameters are fairly choosen. Do you use individual hyperparameters per task/dataset for your method and fixed hyper-parameters for the baselines (or where the hyperparameters there also adapted per task). Did you try to optimize hyperparameters of the baseline?

**Ethical Concerns:**

["NO or VERY MINOR ethics concerns only"]

**Final Justification:**

Insightful new algorithm that outperforms ADAM and is based on Kalman filtering. The performance improvement is rather small for some tasks, thats why I won't rate it as a 5.

**Limitations:**

yes

**Quality:**

3

**Strengths And Weaknesses:**

Strengths:
- Sound approach to gradient descent
- The idea of estimating a low-dimensional covariance in the sub-space of the direction of the current gradient is interesting
- The evaluations are exhaustive and convincing. solid improvement over SOTA in most tasks

Weaknesses:
- Some motivations are unclear (see questions)
- Many results do not contain confidence intervals
- for the experiments with the learning rate schedule, it seems the benefit only comes after the last reduction of the learning rate. Even then, the benefit is rather small.
- It is unclear how much the reported benefits can be attributed to better chosen hyper-parameters.
- Computation time per epoch should be reported for all experiments

---

> ### Author Rebuttal · Authors · 2025-07-31
>
> We sincerely thank the reviewer for their time and thoughtful evaluation of our work. We greatly appreciate the reviewer’s positive feedback on our experimental evaluation of KOALA++, and we are pleased that they found it extensive and convincing.
>
> Below, we address the specific concerns raised:
>
> ---
>
> #### **Clarification on Covariance Approximation**
>
> The reviewer mentions in the paper summary that KOALA++ employs a rank-1 approximation of the covariance matrix. We would like to kindly clarify that the approximation of $P_{k-2}$ in equation (9) is, in fact, rank-2, as it is based on both $v_{k - 1}$ and $H_{k - 1}$, which are generally non-collinear vectors (see equation (15)). This implies that the rank of the implicitly tracked covariance is at least 2.
>
> ---
>
> #### **Missing Confidence Intervals**
>
> Thank you for bringing this to our attention. As noted in lines 181–182 and 221–223, we ran each experiment multiple times and reported average performance metrics. In Table 4, we also reported standard deviations. However, confidence intervals are indeed missing in Tables 2 and 3. In some cases, the values were sourced from the respective original papers, which did not provide confidence intervals. We will include them wherever possible in the revised manuscript.
>
> ---
>
> #### **The Benefit Comes Only After the Final Learning Rate Reduction**
>
> Thank you for the thoughtful comment. We acknowledge that the strongest gains from KOALA++ occur after the final learning rate drop. Our intuition behind this is based on the fact that in the earlier phases of training, the optimization landscape tends to be more non-convex and noisy, where agressive updates from first-order methods are often sufficient or even preferrable. However, as training progresses and the loss surface becomes smoother (after the last learning rate drop), exploiting the curvature of the loss through modelling the uncertainty of the updates in KOALA++ becomes most effective and leads to better generalization. This behavior deserves further investigation.
>
> ___
>
> #### **Fairness of Hyperparameter Tuning**
>
> In most cases, we reported performance numbers directly from prior work, assuming that hyperparameters had been appropriately tuned in those original studies. For baselines where we conducted our own experiments, we performed hyperparameter tuning via grid search. Notice however that in Table 2, 3 and 4 we used the same hyperparameter settings for all the different CNN-based model architectures and datasets. Nonetheless, KOALA++ is consistently the top or the second top performer in all cases. We thus attribute this result more to the method than to the tuning. We will clarify this in the revised the manuscript to make our evaluation more explicit.
>
> ---
>
> #### **Computation Time per Epoch**
>
> We reported the average training time per epoch for the NLP experiment on WikiText-2 to demonstrate the computational efficiency of KOALA++, particularly in comparison with second-order methods such as AdaHessian. Since the computational complexity of our method is task-independent, we felt one representative example would suffice. However, we include the training times for other tasks here below.
>
> ##### **ResNet‑50 (Batch Size = 256)**
>
> | Optimizer         | Time per epoch (s) | FLOPs (class) | Peak Mem (GB) |
> |------------------:|-----------------:|:--------------|--------------:|
> | Adam              | 11.08            | O(N)          | 10.51         |
> | AdaFisher         | 11.64            | O(N)          | 11.91         |
> | AdaHessian        | 63.43            |          | 22.79         |
> | K‑FAC             | 17.16            |       | 12.34         |
> | Shampoo           | 134.63           | O(N^1.5)      | 10.41         |
> | **KOALA++ (ours)**| **13.23**        | **O(N)**      | **10.27**     |
>
>
>
> ##### **Swin Transformer Tiny(Batch Size = 256)**
>
> | Optimizer         | Time per epoch (s) | FLOPs (class) | Peak Mem (GB) |
> |------------------:|-----------------:|:--------------|--------------:|
> | AdamW          | 16.35            | O(N)          | 3.46          |
> | AdaFisher         | 12.00            | O(N)          | 4.38          |
> | AdaHessian        | 42.00            |           | 5.97          |
> | **KOALA++ (ours)**| **16.10**        | **O(N)**      | **4.50**      |
>
>
> ---
>
> #### **Frobenius Norm Justification**
>
> Thank you for the insightful suggestion. Our choice to minimize the Frobenius norm of $P_{k-2}$ was motivated by two considerations: (1) it leads to a convex optimization problem with a unique solution, thereby avoiding ambiguity in the covariance estimate; and (2) it helps keep the estimate bounded, which is important for the numerical stability of our method. That said, we appreciate the reviewer’s proposal to exploit alternative priors, such as minimizing distance to a scaled identity matrix. We will explore this direction in future work.
>
> ---
>
> Please let us know if further clarification is needed. We once again thank the reviewer for their valuable feedback and constructive suggestions.

---

> > ### Comment · Reviewer_csNz · 2025-08-05
> >
> > Thanks for the detailed clarifications, they addressed most concerns. I will stay with my positive evaluation of the paper.

---

> > > ### Author Response · Authors · 2025-08-08
> > > **Response to the reviewer**
> > >
> > > Thank you very much for your thoughtful feedback and for taking the time to engage with our rebuttal. We're glad to hear that the clarifications addressed most of your concerns.
> > >
> > > If you feel that the response resolved the main issues you raised, we would be grateful if you could consider reflecting that in your final assessment of the paper

---

### Note · Authors · 2025-08-15

We thank the reviewers for their valuable feedback.

Three reviewers remained positive after the discussion phase. Reviewers **csNz** and **oqam** explicitly maintained their positive ratings, highlighting the strong empirical results and practical usefulness; reviewer **esMW** had originally a positive score but did not engage in the discussion. Thus, we assume that he/she is satisfied with our clarifications in the rebuttal and in the other discussions, and has not changed the original rating.

We also thank **RxbJ** for raising valuable concerns:

• **Theoretical stability**: Empirically, KOALA++ converges reliably, and Appendix B shows stable spectra even without PSD constraints. Currently, we plan to develop a convergence analysis for KOALA++ as future work, but if we obtain it before the final version of this paper, we would be happy to include it in the appendix.
• **Early-stage convergence**: As we pointed out, performance varies with learning rate schedules; for example, under the cosine decay, KOALA++ often leads already early on. We will include training curves as asked to better understand these behaviors.
• **Related work**: We will clarify distinctions from diagonal + low-rank methods (e.g., SLANG) and EKF-based optimizers. KOALA++ assumes no explicit structure on the covariance, using directional projection and minimal-norm fitting. We will clarify better our motivation.

In terms of efficiency, KOALA++ achieves runtime and memory close to Adam(W) across ResNet, Swin-T, and GPT-1, and is significantly faster than Shampoo and AdaHessian. Its overhead over SGD approaches 1× as batch size grows. We note that all reviewers acknowledged the strength of our empirical results: **csNz** called them “exhaustive and convincing,” **oqam** appreciated “strong empirical results across diverse tasks and architectures,” and even **RxbJ** remarked that “empirical performance looks good.” We believe this supports KOALA++’s generality.

We will update the paper to:
– Unify presentation of FLOPs, memory, and runtime,
– Expand PSD/stability discussion,
– Improve related work coverage,
– Provide guidance for hyperparameters.

We believe KOALA++ provides a practical and scalable (as a quasi-second order method) optimizer, and respectfully ask the AC to consider the reviewers’ consensus and our focused clarifications in their decision.

---

### Decision · Program_Chairs · 2025-09-17

**Decision:**

Accept (poster)

**Comment:**

The paper introduces a new Kalman filter based gradient descent methods that extends a current SOTA method called KOALA. The reviewers highlight the strong empirical results and practical usefulness. But a reviewer argues that the performance improvement is rather small for some tasks.  The central issue for a reviewer was the lack of empirical or theoretical support for the claim that KOALA++’s curvature-aware component only provides benefits late in training. While the authors now attribute differences in convergence profiles to entanglement with learning rate schedulers, this remains a hypothesis without direct experimental validation or comparison to methods like Muon and Shampoo. In summary, this is a borderline paper.